# Diversifying Policy Behaviors via Extrinsic Behavior Curiosity

**Zhenglin Wan** [* 1]  **Xingrui Yu** [* 2 3]  **David Mark Bossens** [2 3]  **Yueming Lyu** [2 3]  **Qing Guo** [2 3]  **Flint Xiaofeng Fan** [2 3]
**Yew-Soon Ong** [2 3 4]  **Ivor W. Tsang** [2 3 4]

## Abstract

Imitation learning (IL) has shown promise in various applications (e.g. robot locomotion) but is often limited to learning a single expert policy, constraining behavior diversity and robustness in unpredictable real-world scenarios. To address this, we introduce Quality Diversity Inverse Reinforcement Learning (QD-IRL), a novel framework that integrates quality-diversity optimization with IRL methods, enabling agents to learn diverse behaviors from limited demonstrations. This work introduces Extrinsic Behavioral Curiosity (EBC), which allows agents to receive additional curiosity rewards from an external critic based on how novel the behaviors are with respect to a large behavioral archive. To validate the effectiveness of EBC in exploring diverse behaviors, we evaluate our method on multiple robot loco-motion tasks. EBC improves the performance of QD-IRL instances with GAIL, VAIL, and DiffAIL across all included environments by up to 185%, 42%, and 150%, even surpassing expert performance by 20% in Humanoid. Furthermore, we demonstrate that EBC is applicable to Gradient-Arborescence-based Quality Diversity Reinforcement Learning (QD-RL) algorithms, where it substantially improves performance and provides a generic technique for learning behavioral diverse policies. The source code of this work is provided at https://github.com/vanzll/EBC.

## 1. Introduction

Imitation learning (IL) enables intelligent systems to quickly learn complex tasks by learning from demonstrations, which is particularly useful when manually designing a reward function is difficult. IL has been applied to many real-world scenarios such as autonomous driving (Bojarski, 2016), robotic manipulation (Zhu et al., 2018), surgical skill learning (Gao et al., 2014), and drone control (Ross et al., 2013). The concept of IL relies on the idea that experts can showcase desired behaviors, when they are unable to directly code them into a pre-defined program. This makes IL applicable to any system requiring autonomous behavior that mirrors expertise (Zare et al., 2024). IL can be divided into two paradigms: behavior cloning (BC), which mimics expert's exact state-action pairs via supervised-learning, and inverse reinforcement learning (IRL), which infers a reward function from expert demonstration and uses reinforcement learning (RL) to learn the policy.

However, traditional IL methods tend to replicate only the specific strategies demonstrated by the expert. If the expert demonstrations cover a narrow range of scenarios, the IL may struggle when faced with new or unseen situations. Additionally, IL faces challenges in stochastic environments where outcomes are uncertain or highly variable. Since the expert's actions may not capture all possible states or contingencies, IL often struggles to learn an optimal strategy for every scenario (Zare et al., 2024). These limitations are further exacerbated when the demonstration data is limited, as the IL algorithm will only learn specific expert behavior patterns. Hence, traditional IL methods are significantly constrained due to the lack of ability to learn diverse behavior patterns to adapt to stochastic and dynamic environments. Moreover, since BC is much more susceptive to the aforementioned challenges (Ross et al., 2011), this work will mainly focus on IRL as our IL paradigm.

On the other hand, the Quality Diversity (QD) algorithm is an emerging optimization paradigm that designed to find a set of diverse solutions to optimization problems while maximizing each solution's fitness value (fitness refers to the problem's objective) (Pugh et al., 2016). For instance, QD algorithms can generate diverse human faces resembling "Elon Musk" with various features, such as different

---
[*]Equal contribution ¹School of Data Science, The Chinese University of Hong Kong, Shenzhen, China ²CFAR, Agency for Science, Technology and Research, Singapore ³IHPC, Agency for Science, Technology and Research, Singapore ⁴College of Computing and Data Science, Nanyang Technological University (NTU), Singapore. Correspondence to: Xingrui Yu <yu_xingrui@cfar.a-star.edu.sg>.

eye colors (Fontaine & Nikolaidis, 2021a). In robot control, the diverse behaviors trained by QD algorithms excels allows adaptation in changing environments (Tjanaka et al., 2022). For example, if an agent's leg is damaged, it can adapt by switching to a policy that uses the other undamaged leg to hop forward due to the diverse behaviors stored in the QD archives (Fontaine & Nikolaidis, 2021a). Naturally, one valuable question is raised: *can we design a novel IRL framework that can combine the respective strengths of traditional IRL and QD algorithms, enabling the agent to learn a broad set of high-performing policies from limited demonstrations?* This framework, which we call Quality Diversity Inverse Reinforcement Learning (QD-IRL), leads to a key challenge of combining QD with gradient-based policy optimization methods: the gradient-based policy optimization methods (such as Proximal Policy Optimization (Schulman et al., 2017)) tend to converge to the policies with high quality, i.e. high cumulative reward, while ignoring behavioral diversity. While the challenge exists for QD-RL too, it will be exacerbated in QD-IRL since the reward function learned by IRL methods will be much more localized, largely inhibiting balanced exploration (Yu et al., 2025).

To address this challenge, we introduce Extrinsic Behavioral Curiosity (EBC). Inspired by the concept of curiosity in psychology, where humans seek out novel and diverse experiences for the sake of learning, EBC encourages exploring diverse behavioral patterns. Specifically, when the agent identifies a new behavior pattern, a reward bonus will be added to the current reward function. Due to the nature of gradient-based policy optimization, the reward bonus will prevent stagnation at local optima and promote balanced exploration. After one sub-region of behavioral patterns is unlocked, the reward bonus of this region is reduced to 0 for further exploitation. Distinct from Intrinsic Curiosity Module (ICM) (Pathak et al., 2017), EBC adopts a higher-level curiosity, owing its name to two defining features. First, EBC promotes the exploration of unseen behavior patterns rather than unseen states. Second, the curiosity reward comes from an extrinsic source; that is, our approach allows an external critic to judge whether or not a behavior is novel or not based on a QD archive stored externally to the agent. Similar ideas have been explored in the work of (Yu et al., 2025), which employs a reward bonus to foster behavioral exploration. However, whereas Yu et al.'s work compute a *single–step* bonus for each $(s, a)$ pair, our work introduces an *episode–level* bonus which aligns closely with episode-based measure computation. To validate our framework, we conduct experiments with limited expert demonstrations across various environments. Notably, our framework is potentially capable of enhancing any IRL method for QD tasks. It reaches or even surpasses expert performance in terms of both QD-Score and coverage in the Walker2d and challenging Humanoid environments. Meanwhile, we empirically

prove that EBC can similarly be applied in QD-RL, significantly improving the QD performance by simply adding the bonus into the reward function. We summarize our contributions as follows:

- We propose a QD-IRL framework, combining the advantages of QD and IRL while addressing the key limitation of IL for various applications.

- We identify the key challenges of combining QD and gradient based policy optimization algorithms. To address this challenge, we introduced Extrinsic Behavioral Curiosity (EBC) to encourage behavior-level exploration, and theoretically proved the effectiveness of our method.

- Our framework can potentially enhance any IRL application that requires learning diverse policies, and can also significantly improve existing SOTA QD-RL algorithm, thus providing a generic framework for future research of learning behavioral-diverse policies.

## 2. Preliminaries and Related Work

### 2.1. Quality Diversity Reinforcement Learning

Distinct from traditional policy optimization which aims to find a single policy to maximize the cumulative reward, Quality Diversity Reinforcement Learning (QD-RL) aims to find **a set of high-performing and diverse policies**. Parametrizing policies based on $\theta \in \Theta \subset \mathbb{R}^n$, the objective function (i.e., fitness) is defined as $f : \Theta \to \mathbb{R}$, which is the cumulative reward in RL context. The diversity is based on a $k$-dimensional measure function $m : \mathbb{R}^n \to \mathbb{R}^k$. The goal is to find policies $\theta \in \Theta$ for each local region in the behavior space $B = \mathbf{m}(\Theta)$. In grid-based algorithms MAP-Elites, one discretizes $B$ into $M$ cells (local regions) and forms a policy archive $\mathscr{A}$, where each cell $i = 1, \dots, M$ represents a small hypercube $[\mathbf{a_i}, \mathbf{b_i}]$ within a multi-dimensional grid of the behavior space. By filling new cells and replacing existing solutions if they are outperformed, the algorithm gradually evolves an archive of high-quality and diverse policies (Chatzilygeroudis et al., 2021; Pugh et al., 2016; Mouret & Clune, 2015a;b). Based on MAP-Elites, Co- variance Matrix Adaptation MAP-Elites (CMA-ME) (Fontaine et al., 2020) further adopts covariance matrix adaptation as the evolution algorithm to find new high-performing solutions more effectively and efficiently. In such algorithms, QD-RL can be treated equivalently to solving $M$ policy-optimization problems of the form

$$
\begin{aligned}
\max_{\theta} \quad & f(\theta) \\
\text{s.t.} \quad & \mathbf{m}(\theta) \in [\mathbf{a_i}, \mathbf{b_i}],
\end{aligned}
\tag{1}
$$

where $i \in \{1, 2, \dots, M\}$ and each constraint corresponds to one cell in the behavior space. Because traditional policy-gradient algorithms will simply optimize the policy towards

the highest cumulative reward, they ignore valuable behavioral diversity. This problem is also referred to as the "behavioral overfitting" issue (Yu et al., 2025), which is one of the key challenges to QD-RL.

The majority of QD-RL works focus on the paradigm of Differentiable Quality Diversity (DQD), which makes use of policy gradient techniques to guide the search for quality diversity. One seminal algorithm in this regard is the Covariance Matrix Adaptation MAP-Elites via a Gradient Arborescence (CMA-MEGA; (Fontaine & Nikolaidis, 2021b)), which maximizes the QD objective

$$g(\boldsymbol{\theta}) = c_0 f(\boldsymbol{\theta}) + \sum_{j=1}^{k} c_j m_j(\boldsymbol{\theta}) \qquad (2)$$

via gradient ascent, where $c_0 \geq 0$ and $c_j \in \mathbb{R}$ for all $j$, where $f$ is the fitness (i.e. the cumulative reward) and $m_j$ are the measure functions. The coefficient vector $\mathbf{c}$ is sampled from a distribution, which is optimized adaptively by the evolutionary strategy (ES) algorithm CMA-ES to optimize the archive improvement, which is based on the total sum of fitness in the archive, where the fitness refers to cumulative reward of the policy.

In CMA-MEGA, the gradients are assumed to be analytically given, i.e. the measures are directly differentiable as a function, which is a limitation. Several works in DQD approximate the gradients. Policy Gradient Assisted (PGA) MAP-Elites (Nilsson & Cully, 2021) incorporates TD3, which is suitable for off-policy learning based on deterministic policy gradient of the fitness. QD Policy Gradient (QD-PG) (Pierrot et al., 2021) introduces the diversity gradient, which uses both quality and diversity critics in a TD3 algorithm, where the diversity reward is based on the distance between state measures with similar policy parameters. Unfortunately, it is a challenge that this state measure is not directly related to the behavior, which can be based on the episode or directly on the parameters. The state-of-the-art QD-RL algorithm, Proximal Policy Gradient Arborescence (PPGA), leverages a vectorized PPO (VPPO) architecture to parallelize gradient approximation: the objective function is assigned a dedicated thread, while each measure function operates on its own thread, enabling efficient parallelized computation of both objectives and measures (Batra et al., 2023). PPGA introduces Markovian Measure Proxy (MMP), a state-based surrogate measure function that correlates strongly with the original measure and allows gradient approximation via policy gradient by treating it as a reward function. It approximates the gradients of both the objective and the $k$ measure functions by subtracting the policy parameters before and after multiple PPO updates. A further modification is its use of exponential evolution strategies (xNES) (Glasmachers et al., 2010), instead of CMA-ES, for updating the search policy, which has more favorable properties for non-stationary problems such as the QD objective (Eq. 2).

Concise paragraphs struggle to adequately summarize the extensive QD-family, and understanding the rationale behind CMA-MEGA and PPGA requires effort. We recommend readers to explore prior works in depth (Batra et al., 2023; Fontaine & Nikolaidis, 2021a) or refer to Appendix A.2 and A.1 for further algorithmic details.

## 2.2. Imitation Learning

In Imitation learning (IL) (Zare et al., 2024), an agent learns high-performing policies from demonstration data. A traditional approach to solve this challenge is Behavior Cloning (BC), which uses supervised learning to learn the policy from demonstrations, a technique which unfortunately suffers from severe error accumulation (Ross et al., 2011). More recent techniques include inverse reinforcement learning (IRL), where one seeks to learn a reward function from the demonstrations and then use RL to train a policy based on that reward function (Abbeel & Ng, 2004).

Early IRL methods estimate rewards using the principle of maximum entropy (Ziebart et al., 2008; Wulfmeier et al., 2015; Finn et al., 2016). Recent adversarial IL methods treat IRL as a distribution-matching problem. For instance, Generative Adversarial Imitation Learning (GAIL) (Ho & Ermon, 2016) trains a discriminator to differentiate between the state-action distribution of the demonstrations and the state-action distribution induced by the agent's policy, and output a reward to guide policy improvement. Improving on GAIL, Variational Adversarial Imitation Learning (VAIL) (Peng et al., 2018) applies a variational information bottleneck (VIB) (Alemi et al., 2016) to the discriminator, improving the stability of adversarial learning. Another technique for adversarial IL is Adversarial Inverse Reinforcement Learning (AIRL) (Fu et al., 2017), which learns a robust reward function by training the discriminator via logistic regression to distinguish expert data from policy data. Generative Intrinsic Reward-driven Imitation Learning (GIRIL) (Yu et al., 2020) computes offline rewards by pretraining a reward model using a conditional VAE (Sohn et al., 2015), combining action encoding with forward dynamics, and has shown superior performance with limited demonstrations. Most recently, DiffAIL (Wang et al., 2024) adopts the loss function of a diffusion model as a distribution matching technique, enhancing the discriminator's performance and generalization capabilities.

**Quality-Diversity Inverse Reinforcement Learning (QD-IRL)** follows the definition of QD-RL, optimizing the same objective, namely the cumulative reward. The key difference is that the true reward function is not accessible, and we infer rewards from the reward model $\mathscr{R}$ trained with the expert demonstration $\mathscr{D}$.

While existing work on Multi-domain Imitation Learning (Seyed Ghasemipour et al., 2019; Yu et al., 2019; Chen et al., 2023) aims to learn context-conditioned policies that generalize across tasks, our approach differs in that we focus on constructing a large archive of policies that represent diverse behaviors. Similarly, skill or option discovery methods (Zhang et al., 2021; Kipf et al., 2019; Jiang et al., 2022) also aim to "find diverse solutions" for policies exhibiting varied behaviors. However, these methods typically generate low-level policies for use within a higher-level MDP (e.g., an MDP with options), often linked to hierarchical RL, where low-level policies are triggered and terminated by specific sub-goals. In contrast, our goal is not to create policies for a hierarchical setup, but rather to build a diverse and high-performing set of policies independently of any hierarchical structure.

## 3. Extrinsic Behavioral Curiosity for QD-IRL

In this section, we will introduce Extrinsic Behavioral Curiosity (EBC), which provides the QD-IRL agent with a curiosity-driven reward bonus to encourage behavior space (a.k.a. policy archive) exploration and address the behavioral-overfitting issue of QD-IRL.

We implement EBC on three different IRL methods, namely GAIL, VAIL, DiffAIL. These IRL methods will learn a reward function from expert demonstration, and the QD-RL algorithm PPGA (Batra et al., 2023) is then applied on this learned reward to train the policy archive. Notably, EBC could be easily integrated into various QD-IRL algorithms and QD-RL algorithm PPGA by simply adding the reward bonus into the learned reward function or true reward function. Figure 1 shows the workflow of our framework. We first sample episodes to form the current search policy. For QD-IRL, we infer rewards from the reward model $\mathscr{R}$ pre-trained using the sampled episodes and expert demonstrations. While all of the optimisation is being done based on imitation rewards, as in (Yu et al., 2025), the solutions are added to the archive based on oracle fitness evaluations to avoid the additional layer of complexity introduced by accumulating effects of fitness estimation errors. For QD-RL, we always use the true rewards from the environment. Then in the third step, we compare the archive and use the measures of episodes to calculate the EBC reward bonus. The orange region of current archive indicates the occupied cells, and the white region indicates the empty cells. If the episode measure occupies the empty cell, the EBC reward bonus $r_{\text{EBC}}$ will be applied to each step of this episode. Then VPPO uses the reward values after applying bonus to approximate gradients for the objective and measures. Subsequently, these gradients are used to produce new solutions, update the archive, update the search distribution, and the search policy based on the CMA-MAEGA paradigm.

Figure 1: Flow diagram of QD-IRL or QD-RL with EBC. The $h$ in step 4 means the cumulative EBC reward.

---

**Algorithm 1** QD-IRL with PPGA as backbone

---

1: **Input:** Initial policy $\theta_0$, VPPO instance to approximate $\nabla g$, $\nabla \mathbf{m}$ and move the search policy, number of QD iterations $N_Q$, number of VPPO iterations to estimate the objective-measure functions and gradients $N_1$, number of VPPO iterations to move the search policy $N_2$, branching population size $\lambda$, and an initial step size for xNES $\sigma_g$. Initial reward model $\mathscr{R}$, Expert data $\mathscr{D}$.

2: Initialize the search policy $\theta_\mu = \theta_0$. Initialize NES parameters $\mu, \Sigma = \sigma_g I$

3: **for** iter $\leftarrow 1$ to $N$ **do**

4:     $f, g, \nabla g, \mathbf{m}, \nabla \mathbf{m} \leftarrow$ VPPO.jacobian$(\theta_\mu, \mathscr{R}, \mathbf{m}, N_1)$ ▷ approx grad using $\mathscr{R}$

5:     $\nabla g \leftarrow$ normalize$(\nabla g)$,    $\nabla \mathbf{m} \leftarrow$ normalize$(\nabla \mathbf{m})$

6:     _ $\leftarrow$ update_archive$(\theta_\mu, f, \mathbf{m})$

7:     **for** $i \leftarrow 1$ to $\lambda$ **do**         ▷ branching solutions

8:         $c \sim \mathscr{N}(\mu, \Sigma)$    ▷ sample gradient coefficients

9:         $\nabla_i \leftarrow c_0 \nabla g + \sum_{j=1}^k c_j \nabla \mathbf{m}_j$

10:         $\theta_i' \leftarrow \theta_\mu + \nabla_i$

11:         $f', *, \mathbf{m}', * \leftarrow$ rollout$(\theta_i')$

12:         $\Delta_i \leftarrow$ update_archive$(\theta_i', f', \mathbf{m}')$   ▷ get archive improvement of each solution.

13:     **end for**

14:     Rank gradient coefficients $\nabla_i$ by $\Delta_i$

15:     Adapt xNES parameters $\mu = \mu', \Sigma = \Sigma'$ based on improvement ranking

16:     $g'(\theta_\mu) := c_{\mu,0} g(\theta_\mu) + \sum_{j=1}^k c_{\mu,j} m_j(\theta_\mu)$,  where $c_\mu = \mu'$         ▷ construct EBC-QD objective function

17:     $\theta_\mu' \leftarrow$ VPPO.train$(\theta_\mu, g', N_2, \mathscr{R})$         ▷ walk search policy using reward model $\mathscr{R}$

18:     $\mathscr{R}$.update$(\mathscr{D}, \theta_\mu')$         ▷ update reward model

19:     **if** there is no change in the archive **then**

20:         Restart xNES with $\mu = 0, \Sigma = \sigma_g I$

21:         Set $\theta_\mu$ to a randomly selected existing cell $\theta_i$ from the archive

22:     **end if**

23: **end for**

---

We provide the pseudo-code of the general procedure of QD-IRL with PPGA in Algorithm 1, where different IRL methods differ from the "update reward model" part and

other parts requiring the reward model to calculate learned reward (highlighted in red). Please refer to Appendix B for algorithms for updating the archive (Algorithm 2), updating the reward model, calculating the rewards with the EBC bonus, and computing the gradients for the objective and measures (Algorithm 3).

### 3.1. Derivation of the EBC reward bonus

The objective of PPGA is based on Eq. 2. We observe that the fitness term $f$ heavily influences PPGA's search policy update direction, as archive improvement is primarily driven by $f$. Therefore, PPGA frequently becomes stuck in local regions because $f$ is calculated by cumulative reward and the policy tends to converge in local optima region while lacking exploration of other regions. Hence, PPGA often generates overlapping solutions with only marginal archive improvement. Additionally, a key challenge in QD-IRL is the conflict between imitation learning and diversity. Limited expert demonstrations lead to behaviorally localized and sometimes misleading reward functions, further exacerbating the problem by restricting search policy updates.

To encourage the search policy to find new behavior patterns (i.e., to explore the empty area in the policy archive), we formulate a reward function based on an indicator function over the archive, i.e. whether the behavior cells has already been explored or not. Since policies and environments are often stochastic, Lemma 3.1 provides a probabilistic guarantee for an EBC reward. More specifically, Lemma 3.1 states that using the indicator function $\mathbb{I}(\mathbf{m} \in \mathscr{A}_e)$ on the empty area of the current archive (denoted as $\mathscr{A}_e$) as the reward function in the standard PPO objective steadily increases the probability that the policy generates new behavior measures.

**Lemma 3.1.** *Suppose the reward function is given by* $r(s_t^i, a_t^i) = \mathbb{I}(\mathbf{m}^i \in \mathscr{A}_e)$, *where* $s_t^i$ *and* $a_t^i$ *represent the state and action at time step t of episode i,* $\mathscr{A}_e$ *is the empty area of archive* $\mathscr{A}$, *and* $\mathbb{I}(\mathbf{m}^i \in \mathscr{A}_e)$ *is the indicator function indicating whether the measure of the i'th episode is in* $\mathscr{A}_e$. *Moreover, let* $\theta \in \Theta$ *be a parameter set with uniform prior over* $\Theta$. *Then if one iteration of PPO successfully increases the objective value, the following inequalities hold:*

$$(1): P(\mathbf{m}^i \in \mathscr{A}_e | \theta_{new}) \geq P(\mathbf{m}^i \in \mathscr{A}_e | \theta_{old}) \quad and$$
$$(2): P(\theta_{new} | \mathbf{m}^i \in \mathscr{A}_e) \geq P(\theta_{old} | \mathbf{m}^i \in \mathscr{A}_e),$$

*where* $P(\mathbf{m}^i \in \mathscr{A}_e | \theta)$ *is the likelihood, i.e. the probability under policy* $\pi_\theta$ *that the measure of episode i belongs to the unoccupied area* $\mathscr{A}_e$, *and* $P(\theta | m \in \mathscr{A}_e)$ *is the posterior, i.e. the probability that the policy that generated an episode-based measure* $\mathbf{m}^i \in \mathscr{A}_e$ *is parametrized by* $\theta$.

*Proof.* **(1).** The objective of policy optimization is

$$
\begin{aligned}
h(\theta, \mathscr{A}_e) &= \mathbb{E}_{\tau_i \sim \pi_\theta} \left[ \sum_{t=0}^{T} \gamma^t r(s_t^i, a_t^i) \right] \\
&= \mathbb{E}_{\tau_i \sim \pi_\theta} \left[ \mathbb{I}(\mathbf{m}^i \in \mathscr{A}_e) \cdot \sum_{t=0}^{T} \gamma^t \right] \\
&= \frac{1 - \gamma^{T+1}}{1 - \gamma} \mathbb{E}_{\tau_i \sim \pi_\theta} \left[ \mathbb{I}(\mathbf{m}^i \in \mathscr{A}_e) \right],
\end{aligned}
\tag{3}
$$

where $T$ is the episode length (rollout length). Since PPO is assumed to monotonically improve the policy, optimizing $h(\theta_{\text{old}}, \mathscr{A}_e)$ through multiple rounds of PPO results in a $\theta_{\text{new}}$ with increased objective, i.e. $h(\theta_{\text{new}}, \mathscr{A}_e) > h(\theta_{\text{old}}, \mathscr{A}_e)$. Therefore, we have

$$\mathbb{E}_{\tau_i \sim \pi_{\theta_{\text{new}}}} \left[ \mathbb{I}(\mathbf{m}^i \in \mathscr{A}_e) \right] \geq \mathbb{E}_{\tau_i \sim \pi_{\theta_{\text{old}}}} \left[ \mathbb{I}(\mathbf{m}^i \in \mathscr{A}_e) \right]. \tag{4}$$

Since $\mathbb{E}_{\tau_i \sim \pi_\theta} \left[ \mathbb{I}(\mathbf{m}^i \in \mathscr{A}_e) \right] = P(\mathbf{m}^i \in \mathscr{A}_e | \theta)$ where $\mathbf{m}^i$ is the measure of episode $\tau_i$, it follows that

$$P(\mathbf{m}^i \in \mathscr{A}_e | \theta_{\text{new}}) \geq P(\mathbf{m}^i \in \mathscr{A}_e | \theta_{\text{old}}).$$

**(2).** Based on Bayes' rule with uniform prior on $\theta$, we have

$$P(\theta | \mathbf{m}^i \in \mathscr{A}_e) = \frac{P(\mathbf{m}^i \in \mathscr{A}_e | \theta) P(\theta)}{P(\mathbf{m}^i \in \mathscr{A}_e)} \propto P(\mathbf{m}^i \in \mathscr{A}_e | \theta).$$

Since $P(\theta)$ and $P(\mathbf{m}^i \in \mathscr{A}_e)$ can be treated as constants when $\theta$ changes, we have

$$P(\theta_{\text{new}} | \mathbf{m}^i \in \mathscr{A}_e) \geq P(\theta_{\text{old}} | \mathbf{m}^i \in \mathscr{A}_e).$$

$\square$

Following Lemma 3.1, we define the EBC reward bonus for all time steps of episode $i$ as

$$r_{\text{EBC}}(\mathbf{m}^i, \mathscr{A}_e) := q \mathbb{I}(\mathbf{m}^i \in \mathscr{A}_e), \tag{5}$$

where $\mathscr{A}_e$ is the empty area of the current archive, $\mathbf{m}^i$ is the measure of episode $i$, and the hyperparameter $q$ controls the weight of the EBC reward. Note that the EBC reward is episode-based, i.e. it is calculated at the end of each episode and added to each step of this episode. Essentially, we compute a policy's behavioral curiosity score at the trajectory's end and propagate this score back to each step's reward to credit the actions in that episode.

By combining the EBC reward bonus with the imitation reward, the EBC algorithm adaptively balances exploration and exploitation; once a region in the archive has been explored, the bonus of this region is set to zero such that the emphasis is again more on exploitation.

## 3.2. Synergy of EBC reward bonus with PPGA

We briefly discuss how using the EBC reward bonus synergizes with PPGA. The algorithm (see Algorithm 1) uses an evolutionary strategy (ES) algorithm, namely xNES, as an emitter to adaptively search for diverse and high-performing policies based on a gradient coefficient vector $\mathbf{c}$ sampled from a search distribution $\mathcal{N}(\mu, \Sigma)$.

If we apply a reward bonus to the original reward, the objective of PPGA transforms into:

$$\max_{\theta} \quad |c_0| g(\theta, \mathscr{A}_e) + \sum_{j=1}^{k} c_j m_j(\theta), \qquad (6)$$

where $g(\theta, \mathscr{A}_e) := f(\theta) + h(\theta, \mathscr{A}_e)$ combines the cumulative EBC reward $h(\theta, \mathscr{A}_e)$ ($\mathscr{A}_e$ is the empty area of current archive) and the fitness $f(\theta)$. The aim of the ES algorithm is to guide the search policy towards the highest archive improvement, i.e. increased total fitness value of all policies in the archive (see Algorithm 2 in Appendix B). Firstly, ES samples multiple coefficient vectors $\mathbf{c}$, where each one leads to a new policy (in the sense that when the gradient of $f, h$ and $m_i$ are calculated using VPPO, the coefficients of these gradient determine the policy update). The ES algorithm then ranks sampled coefficient vectors based on the archive improvement of corresponding new policy and updates the search distribution $\mathcal{N}(\mu, \Sigma)$ accordingly. The search policy $\theta_\mu$ is then updated using the mean of the posterior search distribution as gradient coefficient vector.

The search policy then allows the population to explore from a promising location in policy space. First, VPPO computes the gradients for the objective and measures. Then, search distribution samples *iid* gradient coefficient vectors $\mathbf{c}^i \sim \mathcal{N}(\mu, \Sigma)$ to explore the policy space $\theta_i \leftarrow \theta_\mu + \nabla_i$:

$$\nabla_i = c_0^i \nabla_\theta g(\theta_\mu, \mathscr{A}_e) + \sum_{j=1}^{k} c_j^i \nabla_\theta m_j(\theta_\mu).$$

Making use of the gradient over $h$ in this manner allows to factor in the unoccupied areas to define appropriate gradient coefficients. In many cases, the archive improvement is higher when a new cell is occupied than when replacing an existing elite. Therefore, combined with Lemma 3.1, the resulting algorithm rapidly improves the archive, yielding diverse and high-performing behaviors.

## 3.3. QD-IRL instances with EBC bonus

Having defined the EBC bonus and its connection to PPGA, we now discuss three implementations (GAIL-EBC, VAIL-EBC and DiffAIL-EBC), which are based on standard imitation learning methods GAIL, VAIL, and DiffAIL.

### 3.3.1. GAIL-EBC

Using GAIL (Ho & Ermon, 2016) as a backbone, GAIL-EBC trains a discriminator $D_\psi$ according to

$$\min_{\psi} \max_{\beta \geq 0} \; \mathbb{E}_{(s,a)\sim\mathscr{D}}\Big[\log(-D_\psi(s,a))\Big] \\ + \mathbb{E}_{(s,a)\sim\pi}\Big[\log(1 - D_\psi(s,a))\Big], \qquad (7)$$

where $\mathscr{D}$ is the expert dataset and $\pi$ is the learned policy. The reward function for GAIL-EBC is given by

$$r(s_t^i, a_t^i, \mathbf{m}^i, \mathscr{A}_e) = -\log\left(1 - D_\psi(s_t^i, a_t^i)\right) + r_{\text{EBC}}(\mathbf{m}^i, \mathscr{A}_e), \qquad (8)$$

where $\mathscr{A}_e$ is the empty region of the current policy archive.

### 3.3.2. VAIL-EBC

VAIL (Peng et al., 2018) modifies GAIL by using a variational information bottleneck to the discriminator. Using VAIL as a backbone, VAIL-EBC trains a discriminator $D_\psi$ according to

$$\min_{D_\psi, E'} \max_{\beta \geq 0} \mathbb{E}_{(s,a)\sim\mathscr{D}}\Big[\mathbb{E}_{z\sim E'(z|s,a)}\big[\log(-D_\psi(z))\big]\Big] \\ + \mathbb{E}_{(s,a)\sim\pi}\Big[\mathbb{E}_{z\sim E'(z|s,a)}\big[-\log(1 - D_\psi(z))\big]\Big] \\ + \beta \mathbb{E}_{s\sim\tilde{\pi}}\big[d_{KL}(E'(z|s,a)||p(z)) - I_c\big], \qquad (9)$$

where $\tilde{\pi}$ is the mixture of expert policy and agent policy, and $E'$ is the latent variable encoder. The reward function for VAIL-EBC is given by

$$r(s_t^i, a_t^i, \mathbf{m}^i, \mathscr{A}_e) = -\log\left(1 - D_\psi(\boldsymbol{\mu}_{E'}(s_t^i, a_t^i))\right) + r_{\text{EBC}}(\mathbf{m}^i, \mathscr{A}_e), \qquad (10)$$

where $\boldsymbol{\mu}_{E'}(s_t^i, a_t^i)$ represents the mean of the encoded latent variable distribution.

### 3.3.3. DiffAIL-EBC

DiffAIL (Wang et al., 2024) is a state-of-the-art adversarial IL method which adopts the distribution-matching capability of diffusion models to enhance the discriminator. The objective for training the discriminator is based on the same principle as GAIL, but incorporates diffusion noise for a more robust discrimination:

$$\min_{\pi_\theta} \max_{D_\psi} \; \mathbb{E}_{\varepsilon\sim\mathcal{N}(0,I), t\sim\mu(1,T), x=(s,a)\sim\mathscr{D}}\Big[\mathbb{E}_{x\sim\pi_e}\big[\log(D_\psi(x,\varepsilon,t))\big] \\ + \mathbb{E}_{x\sim\pi_\theta}\big[\log(1 - D_\psi(x,\varepsilon,t))\big]\Big], \qquad (11)$$

where $D_\psi(x, \varepsilon, t)$ is the discriminator function at time $t$ and $\varepsilon \sim \mathcal{N}(0, I)$. The specific formulation of $D_\psi$ is given by

$$D_\psi(x_t, \varepsilon, t) = \exp\left(-\text{Diff}_\psi(x_t, \varepsilon, t)\right) \\ = \exp\left(-||\varepsilon - \varepsilon_\psi(\sqrt{\alpha_i} x_t + \sqrt{1 - \alpha_t}\varepsilon, t)||^2\right), \qquad (12)$$

where $\varepsilon_\psi$ is a model parameterized by $\psi$ to predict $\varepsilon$ from $x_t$ and $t$. This follows DDPM (Ho et al., 2020), where rather than approximating the process mean, the process noise is predicted. This is based on a reparametrization trick which represents the forward process as $x_t = \sqrt{\alpha_t}x_0 + \sqrt{1-\alpha_t}\varepsilon$, where $\alpha_t = \prod_{t'=1}^{t}(1-\beta_{t'})$ is the variance schedule $\beta_t$. The reward function for DiffAIL-EBC is:

$$r(s_t^i, a_t^i, \mathbf{m}^i, \mathscr{A}_e) = -\frac{1}{T}\sum_{t=1}^{T}\log\left(1 - D_\psi\left((s_t^i, a_t^i), \varepsilon, t\right)\right) + r_{\text{EBC}}(\mathbf{m}^i, \mathscr{A}_e).$$

(13)

## 4. Experiments

### 4.1. Learning Diverse High-Performing Skills from Limited Demonstrations

**Experiment Setup** We evaluate our framework on three popular MuJoCo (Todorov et al., 2012) environments: Halfcheetah, Humanoid, and Walker2d. The goal in each task is to maximize forward progress and robot stability while minimizing energy consumption, and the measure function maps the policy into a vector where each dimension indicates the proportion of time a leg touches the ground. We evaluate using four common QD-RL metrics: 1) **QD-Score**, the sum of scores of all nonempty cells in the archive. QD-Score is the most important metric in QD-IRL as it aligns with the objective of QD-IRL as in equation (1); 2) **Coverage**, the percentage of nonempty cells, indicating the algorithm's ability to discover diverse behaviors; 3) **Best Reward**, the highest score found by the algorithm; and 4) **Average Reward**, the mean score of all nonempty cells, reflecting the ability to discover high-performing policies across the behavior space. We use the true reward functions to calculate these metrics. Please refer to Appendix C for hardware setup and more implementation details.

**Demonstrations** We use a policy archive obtained by PPGA to generate expert demonstrations. In line with a real-world scenario with limited demonstrations, we first sample the top 500 high-performance elites from the archive as a candidate pool, and then select a few demonstrations such that they are as diverse as possible. This process results in 4 diverse demonstrations (episodes) per environment. Appendix D shows the statistical properties for selected demonstrations.

**Overall Performance** As mentioned in section 3, we focus on applying EBC to three IRL methods-GAIL, VAIL and DiffAIL, to compare the performance improvement. Additionally, we also include a few widely-used and state-of-the-art IL methods as baselines: 1) Traditional IRL: Max-Ent IRL, 2) Online reward methods: AIRL, and 3) Data-driven methods: GIRIL. Each baseline learns a reward function, which is then used to train standard PPGA under identical

settings for all baselines. Hyperparameter details are provided in Appendix E. All the experiments are averaged with three random seeds. We also study the effect of hyperparameter $q$, which controls the weight of EBC reward bonus term in Eq. 5. We found $q = 2$ performs the best among $[1, 2, 5]$, and thus use $q = 2$ for GAIL-EBC, VAIL-EBC and DiffAIL-EBC across the experiments. Please refer to Appendix E.3 for details.

Figure 2 compares the training curves across four metrics for three selected IL methods, their respective EBC-improved version, and the expert (PPGA with true reward function). By comparing the line with the same color, key observations include: 1) In Walker2d , EBC consistently improved GAIL, VAIL and DiffAIL in terms of the most crucial QD metric – the QD score – by 38%, 32% and 71%, respectively. In Humanoid, EBC consistently improved GAIL, VAIL and DiffAIL in terms of the most crucial QD metric-QD score by 185%, 42% and 150%, respectively. In Halfcheetah, the performance of DiffAIL and its EBC-improved version is approximately the same, but EBC improves GAIL and VAIL in terms of QD-Score by 30% and 5%, respectively. 2) The coverage metric of all EBC-improved model consistently reaches near 100%, and significantly outperforms other the respective naive version and PPGA with true reward. Notably, PPGA expert with true reward explored less than 50% of the cells in Humanoid. 3) In Humanoid, the performance of VAIL-EBC and DiffAIL-EBC reaches or even surpasses PPGA with true reward in terms of QD-Score by about 20%. It is also notable that the BestReward of EBC variants takes longer to converge, which is normal because the higher the coverage, the longer it will take to evolve the elite solutions for all the covered cells. For detailed quantitative results (including three IL baselines) on the different QD metrics, please refer to Table 8 in Appendix G. We also conducted experiments to report the results using an additional metric named Complementary Cumulative Distribution Function (CCDF) (Vassiliades et al., 2017), please refer to Appendix F for details.

**Policy Archive Visualization** Figure 3 visualizes the policy archives learned by PPGA with true reward (True Reward), GAIL, GAIL-EBC in Halfcheetah and Humanoid. The archive produced by GAIL-EBC covers a much larger area, highlighting the importance of behavior exploration.

**Extension to QD-RL** Surprisingly, we observe that VAIL-EBC achieves near-expert QD-Score and that DiffAIL-EBC even surpasses expert QD-Score in the Humanoid environment in Figure 2. This is counter-intuitive because it is common belief that IL can hardly surpass expert performance. However, one may argue here for the benefit of IRL compared to traditional RL because the imitation reward function may be aligned with the true reward but potentially be easier to learn due to the function being less sparse and

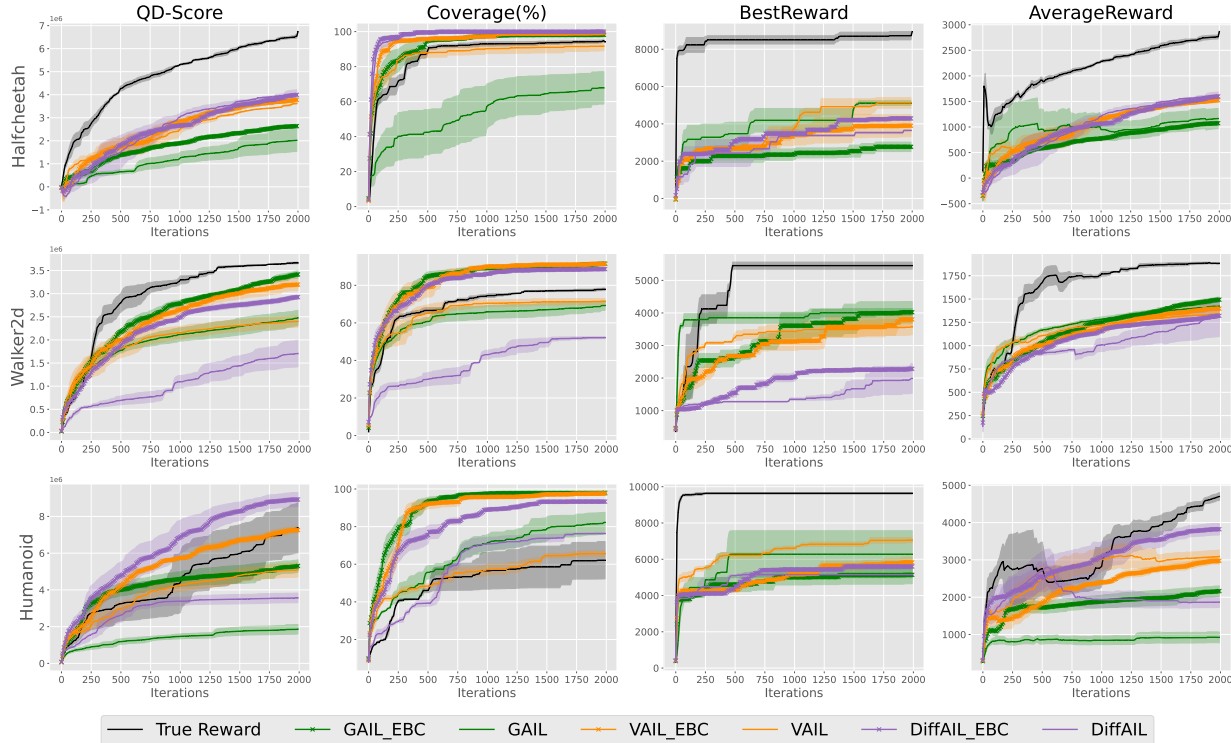

Figure 2: EBC improves performance of QD-IRL instances with GAIL, VAIL and DiffAIL in three robot locomotion task. Lines indicate the mean and shaded areas indicate the standard deviations.

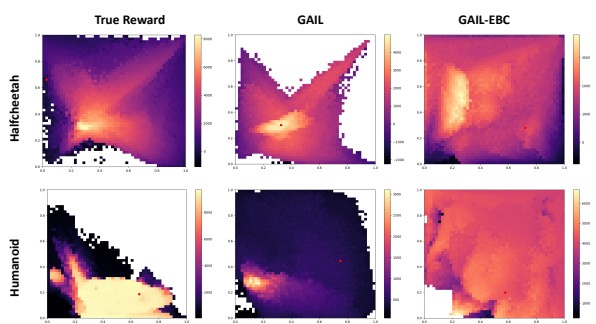

Figure 3: Visualization of well-trained policy archives.

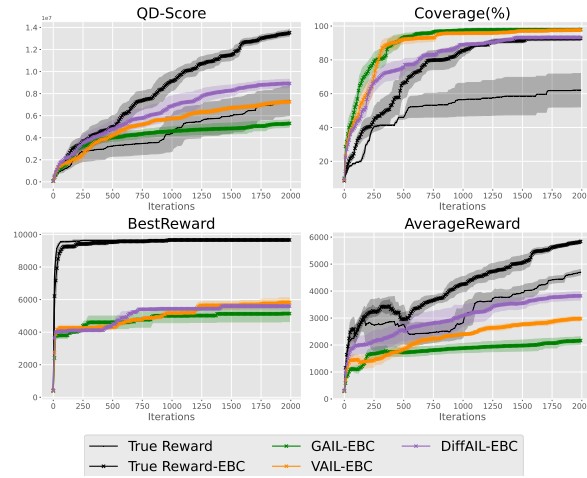

Figure 4: EBC improves PPGA with true reward (True Reward) in the Humanoid environment.

more continuous. To assess this potential benefit, we add our EBC reward bonus to the true reward function of the QD-RL task. Figure 4 shows that the QD performance significantly improves after the EBC reward bonus is applied. The increase in coverage is particularly high, namely from 60% to near 100%, justifying the effectiveness of the curiosity module in exploring diverse policies. PPGA with true reward and EBC bonus significantly outperforms all IRL algorithms on all metrics, with the exception of the coverage.

**Experiments on Various Measure Metrics** To further assess the generality of our framework, we applied EBC to the GAIL baseline (GAIL-EBC) and evaluated it on tasks that

differ in both the *type* and the number of dimension of the QD measure **m**. Specifically, we used: 1) **Jump** ($dim$=1), a scalar that records the height of the *lowest* foot at each step; 2) **Angle** ($dim$=2), a 2-D vector $\phi_t = (\cos\alpha, \ \sin\alpha)^\top$ capturing the yaw orientation $\alpha$ of the torso about the $z$-axis; and 3) **Feet Contact** ($dim$=4), a 4-D vector that stores, for each of the ant's four legs, the fraction of time steps dur-

ing which the leg touches the ground. Table 1 reports that GAIL-EBC improves QD-Score of GAIL throughout every scenario and significantly improve the Coverage of GAIL in humanoid environment.

Table 1: Performance of GAIL versus GAIL-EBC on tasks with measures of increasing dimensionality: *Jump* (1-D), *Angle* (2-D), and *Feet Contact* (4-D). Results are reported as mean ± one standard deviation over three seeds.

| Environment | Measure (*dim*) | Model | QD-Score | Coverage (%) |
|---|---|---|---|---|
| Hopper | Jump (1) | GAIL | 64 017 ± 5 558 | 100.0 ± 0.0 |
| | | GAIL-EBC | **81 987** ± 6 890 | **100.0** ± 0.0 |
| Humanoid | Angle (2) | GAIL | 63 771 ± 7 113 | 53.0 ± 3.0 |
| | | GAIL-EBC | **170 273** ± 19 711 | **96.0** ± 0.0 |
| | Jump (1) | GAIL | 18 812 ± 6 412 | 41.0 ± 31.0 |
| | | GAIL-EBC | **90 631** ± 61 894 | **85.0** ± 15.0 |
| Ant | Feet Contact (4) | GAIL | −484 468 ± 3 264 | 75.84 ± 0.0 |
| | | GAIL-EBC | **−52 521** ± 383 213 | **77.28** ± 3.2 |

**Statistical Analysis** We use Tukey HSD test to evaluate the statistical significance of the QD-Scores (Table 8). Comparing our EBC-enhanced algorithms to their counterparts, 7/9 effects are positive and 5 of these are significant with $p \leq 0.05$. See Appendix H for more details.

**Effects of Number, Quality and Diversity of Demonstrations** Here, we study how the *quantity*, *quality*, and *diversity* of demonstrations affect QD performance in QD-IRL. To isolate these factors we apply the EBC bonus to a GAIL baseline (GAIL-EBC) on the *Humanoid* domain and report QD-Score and coverage in Table 10 (Appendix I). Vanilla GAIL shows no consistent benefit—and sometimes a decline—when more demonstrations are added, whereas GAIL-EBC benefits from a notable improvement as diverse demonstrations are added. Even as few as 2 diverse demonstrations are beneficial to yield an observable QD-Score improvement of about 15%, and not much if any improvement is observed by adding additional demonstrations. Imitation learning with GAIL-EBC from 4 non-diverse demonstrations actually reduces the QD-Score by roughly 30%. These results confirm the importance and scalability of imitation learning from diverse experts using GAIL-EBC. The results for GAIL indicate the trade-off between data quantity (number of demonstrations), as well as the fitness and diversity of those demonstrations.

### 4.2. Few-Shot Adaptation from the Imitation Archive

**Experiment Setup** We evaluate our method on few-shot adaptation scenarios with three types of perturbation (Grillotti et al., 2024). For each task, the reward is the same but the MDP's dynamics is perturbed. In few-shot adaption tasks, no re-training is allowed and we evaluate the top-performing skills for each method while varying the perturbation to measure the robustness of the different algorithms, in particular GAIL-ECB and GAIL. *Humanoid -*

*Hurdles* requires the agent to jump over hurdles of varying heights. *Humanoid - Motor Failure* requires the agent to adapt to different degrees of failure in the motor controlling its left knee. Finally, *Walker2d - Friction* requires the agent to adapt to varying levels of ground friction. Here, we evaluate the agent's ability to adjust its locomotion strategy to a new perturbed MDP. See Appendix J for more details.

**Evaluation Metrics** Following (Grillotti et al., 2024), we report the Inter-Quartile Mean (IQM) value for each metric with the estimated 95% Confidence Interval (CI) (Agarwal et al., 2021).

**Results** We show that we can harness the learned skills from the EBC-enhanced QD-IRL methods to adapt better than or comparable to its unenhanced counterpart in three different sets of perturbations each of which corresponds to 5 unseen MDPs (Fig. 5).

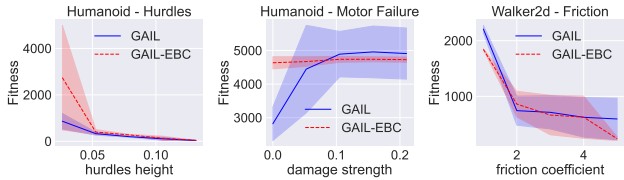

Figure 5: Performance for GAIL-EBC and GAIL in environments with different levels of perturbations after few-shot adaptation. The lines and shaded area represent the IQM and the 95% CI based on exhaustive search over 2 archives.

## 5. Conclusion

In this work, we propose Extrinsic Behavioral Curiosity (EBC), a simple but effective approach which is highly extendable, and theoretically derived the rationale behind it. EBC can be seamlessly integrated into and can potentially improve any IRL method in QD contexts, providing the a generic QD-IRL framework for future research. Our framework follows the paradigm of IRL to learn a QD-enhanced reward function, and uses a QD-RL algorithm to optimize a policy archive. By encouraging behavior-level exploration, our framework addresses the key challenges of QD-IRL. Extensive experiments show that our framework can often improve QD-IRL methods and significantly outperform baselines. Additionally, we empirically prove that EBC can also significantly improve the performance of the state-of-the-art QD-RL algorithm, PPGA, highlighting the sub-optimality of the true reward function in QD-RL settings. With EBC, Gradient-Arborescence based QD-RL and QD-IRL algorithms can learn archives of diverse and high-performing policies more efficiently, pushing the boundary of diverse policy acquisition. We also provide potential limitations of our work in Appendix K and additional discussion in Appendix L.

## Acknowledgements

We thank the anonymous reviewers and area chair for their helpful comments. XY, DB, YL, QG, FXF, YSO and IWT are supported by CFAR, Agency for Science, Technology and Research, Singapore. This research / project is supported by the National Research Foundation, Singapore and Infocomm Media Development Authority under its Trust Tech Funding Initiative, and the National Research Foundation, Singapore under its National Large Language Models Funding Initiative (AISG Award No: AISG-NMLP-2024-004). Any opinions, findings and conclusions or recommendations expressed in this material are those of the author(s) and do not reflect the views of National Research Foundation, Singapore and Infocomm Media Development Authority.

## Impact Statement

This research has the potential to significantly advance the field of robotics by enabling robots to autonomously develop a wide range of adaptive and efficient locomotion strategies. By leveraging extrinsic behavioral curiosity, this work encourages robots to explore and master diverse behaviors in complex, unstructured environments.

The implications of this research are far-reaching. In real-world applications such as search and rescue, planetary exploration, and assistive robotics, robots equipped with this capability can adapt to dynamic and unpredictable conditions, enhancing their utility and reliability. Furthermore, the emphasis on diversity in locomotion behaviors can lead to more robust and generalizable robotic systems, reducing the need for extensive manual tuning and domain-specific engineering.

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

# A. Background

## A.1. Traditional QD Algorithms

Some traditional Quality Diversity optimization methods integrate Evolution Strategies (ES) with MAP-Elites (Mouret & Clune, 2015b), such as Covariance Matrix Adaptation MAP-Elites (CMA-ME) (Fontaine et al., 2020). CMA-ME uses CMA-ES (Hansen & Ostermeier, 2001) as ES algorithm generating new solutions that are inserted into the archive, and uses MAP-Elites to retains the highest-performing solution in each cell. CMA-ES adapts its sampling distribution based on archive improvements from offspring solutions. However, traditional ES faces low sample efficiency, especially for high-dimensional parameters such as neural networks.

Differentiable Quality Diversity (DQD) improves exploration and fitness by leveraging the gradients of both objective and measure functions. Covariance Matrix Adaptation MAP-Elites via Gradient Arborescence (CMA-MEGA) (Fontaine & Nikolaidis, 2021a) optimizes both objective function $f$ and measure functions $\mathbf{m}$ using gradients with respect to policy parameters: $\nabla f = \frac{\partial f}{\partial \theta}$ and $\nabla \mathbf{m} = \left( \frac{\partial m_1}{\partial \theta}, \ldots, \frac{\partial m_k}{\partial \theta} \right)$. The objective of CMA-MEGA is $g(\theta) = |c_0| f(\theta) + \sum_{j=1}^{k} c_j m_j(\theta)$, where the coefficients $c_j$ are sampled from a search distribution. CMA-MEGA maintains a search policy $\pi_{\theta_\mu}$ in policy parameter space, corresponding to a specific cell in the archive. CMA-MEGA generates local gradients by combining gradient vectors with coefficient samples from CMA-ES, creating branched policies $\pi_{\theta_1}, \ldots, \pi_{\theta_\lambda}$. These branched policies are ranked based on their archive improvement, which measures how much they improve the QD-Score (one QD metric, which will be discussed in the experiment section) of the archive. The ranking guides CMA-ES to update the search distribution, and yields a weighted linear recombination of gradients to step the search policy in the direction of greatest archive improvement. Covariance Matrix Adaptation MAP-Annealing via Gradient Arborescence (CMA-MAEGA) (Fontaine & Nikolaidis, 2023), introduces soft archives, which maintain a dynamic threshold $t_e$ for each cell. This threshold is updated by $t_e \leftarrow (1 - \alpha) t_e + \alpha f(\pi_{\theta_i})$ when new policies exceed the cell's threshold, where $\alpha$ balances the time spent on exploring one region before exploring another region. This adaptive mechanism allows more flexible optimization by balancing exploration and exploitation.

## A.2. Details about PPGA and Related Background

To help readers to better understand the background of QD-RL, we begin with Covariance Matrix Adaptation MAP-Elites via a Gradient Arborescence (CMA-MEGA) (Fontaine & Nikolaidis, 2021a).

For a general QD-optimization problem, the objective of CMA-MEGA is:

$$g(\theta) = |c_0| f(\theta) + \sum_{j=1}^{k} c_j m_j(\theta), \tag{14}$$

In this context, $m_j(\theta)$ represents the $j$-th measure of the solution $\theta$, and $k$ is the dimension of the measure space. The objective function of CMA-MEGA is dynamic because the coefficient for each measure, $c_j$, is updated adaptively to encourage diversity in $m$. For instance, if the algorithm has already found many solutions with high $m_1$ values, it may favor new solutions with low $m_1$ values by making $c_1$ negative, thus minimizing $m_1$. However, the coefficient for the fitness function $f$ will always be positive, as the algorithm always seeks to maximize fitness. This objective function ensures that CMA-MEGA simultaneously maximizes fitness $f$ and encourages diversity across the measures $m$. We update $\theta$ by differentiating objective (14) and use gradient descent based optimization approaches, since DQD assumes $f$ and $m$ are differentiable.

Furthermore, the coefficients $c_j$ are sampled from a distribution, which is maintained using Covariance Matrix Adaptation Evolution Strategy (CMA-ES) (Hansen, 2016). Specifically, CMA-ES updates the coefficient distribution by iteratively adapting the mean $\mu$ and covariance matrix $\Sigma$ of the multivariate Gaussian distribution $N(\mu, \Sigma)$, from which the coefficients $c_j$ are sampled. At each iteration, CMA-MEGA ranks the solutions based on their archive improvement (i.e. How much they improve the existing solutions of occupied cell). The top-performing solutions are used to update $\mu$, while $\Sigma$ is adjusted to capture the direction and magnitude of successful steps in the solution space, thereby refining the search distribution over time.

In CMA-MAEGA (Fontaine & Nikolaidis, 2023), the concept of **soft archives** is introduced to improve upon CMA-MEGA. Instead of maintaining the best policy in each cell, the archive employs a dynamic threshold, denoted as $t_e$. This threshold is updated using the following rule whenever a new policy $\pi_{\theta_i}$ surpasses the current threshold of its corresponding cell $e$:

$$t_e \leftarrow (1 - \alpha)t_e + \alpha f(\pi_{\theta_i})$$

Here, $\alpha$ is a hyperparameter called the **archive learning rate**, with $0 \leq \alpha \leq 1$. The value of $\alpha$ controls how much time is spent optimizing within a specific region of the archive before moving to explore a new region. Lower values of $\alpha$ result in slower threshold updates, emphasizing exploration in a particular region, while higher values promote quicker transitions to different areas. The concept of soft archives offers several theoretical and practical advantages, as highlighted in previous studies.

PPGA (Batra et al., 2023) is directly built upon CMA-MAEGA. We summarize the key synergies between PPGA and CMA-MAEGA as follows:(1) In reinforcement learning (RL), the objective functions $f$ and $m$ in Equation 14 are not directly differentiable. To address this, PPGA employs **Markovian Measure Proxies (MMP)**, where a single-step proxy $\delta(s_t)$ is treated as the reward function of an MDP. PPGA utilizes $k+1$ parallel PPO instances to approximate the gradients of $f$ and each measure $m$, where $k$ is the number of measures. Specifically, the gradient for each $i$-MDP is computed as the difference between the parameters $\theta_{i,\text{new}}$ after multi-step PPO optimization and the previous parameters $\theta_{i,\text{old}}$. (2) Once the gradients are approximated, the problem is transformed into a standard DQD problem. PPGA then applies a modified version of CMA-MAEGA to perform quality diversity optimization. The key modifications include:

1. **Replacing CMA-ES with xNES for Stability**: To improve stability in noisy reinforcement learning environments, CMA-ES was replaced with Exponential Natural Evolution Strategy (xNES). While CMA-ES struggled with noisy, high-dimensional tasks due to its cumulative step-size adaptation mechanism, xNES provided more stable updates to the search distribution, especially in low-dimensional objective-measure spaces, and maintained search diversity.

2. **Walking the Search Policy with VPPO**: PPGA "walks" the search policy over multiple steps by optimizing a new multi-objective reward function with VPPO (Vectorized Proximal Policy Optimization). This is done by leveraging the mean gradient coefficient vector from xNES, ensuring stable and controlled movement toward greater archive improvement.

## B. Algorithm Pseudo Code

Algorithm 2 shows the principle of updating the policy archive using MAP-Elites Annealing. Algorithm 3 explaining how our reward model calculates rewards and how the reward model being updated.

---

**Algorithm 2** Update Archive

---

**Input:** Solution $\theta$ to insert, fitness $f$, measures $\mathbf{m} = <m_1, ..., m_k>$, archive $\mathscr{A}$, archive learning rate $\alpha$
$\theta_{inc}, f_{inc} \leftarrow \mathscr{A}[\mathbf{m}]$ if $\mathscr{A}[\mathbf{m}]$ is nonempty else $None, 0$
$\Delta_i = 0$
**if** $f > f_{inc}$ **then**
    insert $\theta$ into cell $\mathscr{A}[\mathbf{m}]$
    $f_{inc} \leftarrow (1 - \alpha)f_{inc} + \alpha f$
    $\Delta_i = f - f_{inc}$
**end if**
**return** $\Delta_i$

---

---

**Algorithm 3** Reward Model $\mathscr{R}$ (using GAIL as the backbone for example)

---

1: **Initialize:** Discriminator $D_\psi$
2:
3: **Method: Reward Calculation for VPPO.compute_jacobian()**
4:    `def get_episode_reward(self, episode, current archive` $\mathscr{A}$`):`
5:      $s_1, a_1, s_2, a_2, \ldots, s_k, a_k \leftarrow episode$
6:      $r_1, r_2, r_3 \ldots, r_k \leftarrow D_\psi([\mathbf{s}, \mathbf{a}])$             $\triangleright$ GAIL batch reward
7:      $m \leftarrow episode.get\_measure()$
8:      $r_{\text{EBC}} \leftarrow q\mathbb{I}(\mathbf{m}^i \in \mathscr{A}_e)$             $\triangleright$ calculate curiosity-driven reward
9:      **For** $i = 1 \rightarrow k$
10:        $r_i \leftarrow r_i + r_{\text{EBC}}$             $\triangleright$ calculate total reward
11:      **return** $r_1, r_2, r_3 \ldots, r_k$
12:
13: **Method: Update reward model**
14:    `def update(self,` $\mathscr{D}, \pi_\theta$`):`
15:      Sample a batch of trajectories $(\mathbf{s}^\pi, \mathbf{a}^\pi)$ from $\pi_\theta$
16:      Update discriminator $D_\psi$ by minimizing:

$$\mathscr{L}_D(\psi) = \mathbb{E}_{(\mathbf{s},\mathbf{a}) \sim \mathscr{D}}[-\log D_\psi(\mathbf{s}, \mathbf{a})] + \mathbb{E}_{(\mathbf{s},\mathbf{a}) \sim \pi_\theta}[-\log(1 - D_\psi(\mathbf{s}, \mathbf{a}))]$$

17:      Repeat until the model converges or the number of epochs is reached
18:      **return** Updated $D_\psi$

---

## C. Hardware Setup

Our experiments are based on the PPGA implementation using the Brax simulator (Freeman et al., 2021), enhanced with QDax wrappers for measure calculation (Lim et al., 2022). We leverage pyribs (Tjanaka et al., 2023) and CleanRL's PPO (Huang et al., 2020) for implementing the PPGA algorithm. The observation space sizes for these environments are 17, 18, and 227, with corresponding action space sizes of 6, 6, and 17. All Experiments are conducted on a system with four A40 48G GPUs, an AMD EPYC 7543P 32-core CPU, and a Linux OS. Each single experiment only requires one A40 48G GPU and takes roughly two days.

## D. Demonstration Details

Figure 6 shows the MuJoCo environments used in our experiments. Table 2 shows the detailed information of the demonstrations in our experiment. Figure 7 visualizes the selected policies in the behavior space.

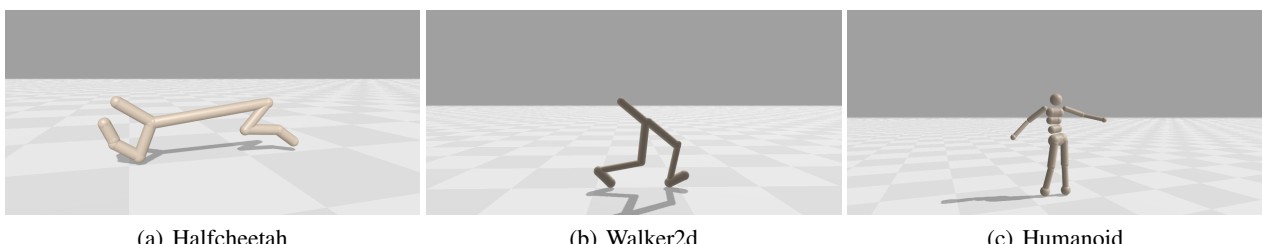

         (a) Halfcheetah                      (b) Walker2d                      (c) Humanoid

Figure 6: MuJoCo Environments.

Table 2: Demonstrations are generated from **top-500** high-performance elites.

| Tasks | Demo number | Attributes | min | max | mean | std |
|-------|-------------|------------|-----|-----|------|-----|
| Halfcheetah | 4 | Length | 1000 | 1000 | 1000.0 | 0.0 |
| | | Demonstration Return | 3766.0 | 8405.4 | 5721.3 | 1927.6 |
| Walker2d | 4 | Length | 356.0 | 1000.0 | 625.8 | 254.4 |
| | | Demonstration Return | 1147.9 | 3721.8 | 2372.3 | 1123.7 |
| Humanoid | 4 | Length | 1000.0 | 1000.0 | 1000.0 | 0.0 |
| | | Demonstration Return | 7806.2 | 9722.6 | 8829.5 | 698.1 |

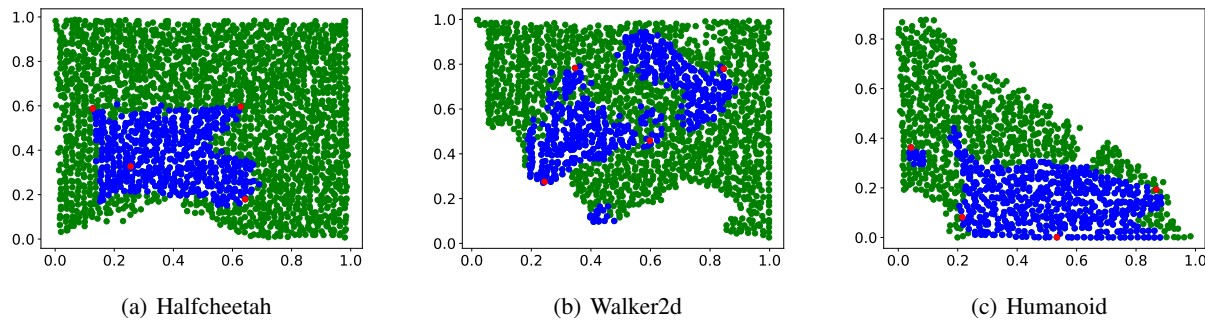

(a) Halfcheetah  (b) Walker2d  (c) Humanoid

Figure 7: Visualization of the behavior space. Green indicates the full expert behavior space, blue indicates the selected top-500 elites, and red indicates the demonstrators. The x axis is the proportion of time Leg 1 touches the ground and the y axis is the proportion of time Leg 2 touches the ground.

# E. Hyperparameter Setting

## E.1. Hyperparameters for PPGA

Table 3 summarizes a list of hyperparameters for PPGA policy updates.

Table 3: List of relevant hyperparameters for PPGA shared across all environments.

| Hyperparameter | Value |
|----------------|-------|
| Actor Network | [128, 128, Action Dim] |
| Critic Network | [256, 256, 1] |
| $N_1$ | 10 |
| $N_2$ | 10 |
| PPO Num Minibatches | 8 |
| PPO Num Epochs | 4 |
| Observation Normalization | True |
| Reward Normalization | True |
| Rollout Length | 128 |
| Grid Size | 50 |
| Env Batch Size | 3,000 |
| Num iterations | 2,000 |

## E.2. Hyperparameters for IL

Table 4 summarizes a list of hyperparameters for AIRL, GAIL.

Table 4: List of relevant hyperparameters for AIRL, GAILs shared across all environments.

| Hyperparameter | Value |
| --- | --- |
| Discriminator | [100, 100, 1] |
| Learning Rate | 3e-4 |
| Discriminator Num Epochs | 1 |

Table 5 summarizes a list of hyperparameters for VAIL.

Table 5: List of relevant hyperparameters for VAILs shared across all environments.

| Hyperparameter | Value |
| --- | --- |
| Discriminator | [100, 100, (1, 50, 50)] |
| Learning Rate | 3e-4 |
| Information Constraint $I_c$ | 0.5 |
| Discriminator Num Epoch | 1 |

Table 6 summarizes a list of hyperparameters for GIRIL.

Table 6: List of relevant hyperparameters for GIRIL shared across all environments.

| Hyperparameter | Value |
| --- | --- |
| Encoder | [100, 100, Action Dim] |
| Decoder | [100, 100, Observation Dim] |
| Learning Rate | 3e-4 |
| Batch Size | 32 |
| Num Pretrain Epochs | 10,000 |

Table 7 shows the hyperparameter setting for DiffAIL and DiffAIL-EBC.

Table 7: List of relevant hyperparameters for DiffAIL and DiffAIL-EBC.

| Hyperparameter | Value |
| --- | --- |
| Variance schedule ($\beta_t$) | linear from $1e-4$ to $2e-2$ |

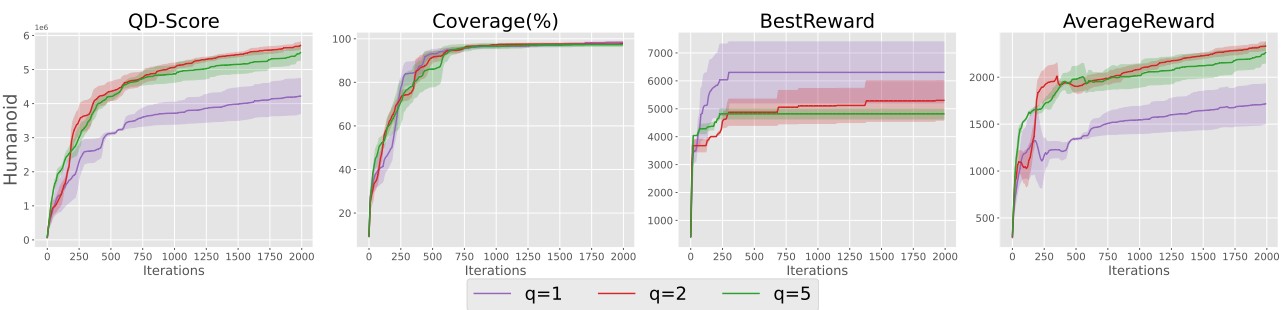

Figure 8: Study of Hyperparameter $q$

### E.3. Hyperparameter Study of $q$

In our work, there is only one hyperparameter $q$ which controls the weight of reward bonus. We select one environment, namely Humanoid, and compare the QD performance between $q$ value 1,2 and 5. Figure 8 shows that $q = 2$ is the best in this study.

However, the best choice of $q$ varies according to the property of different environments, and it is not possible to find the exact optimal $q$ value for each potential environment since $q$ value is continuous. Hence, for simplicity, we select $q = 2$ in our experiments.

## F. Results on Additional Evaluation Metric

The complementary cumulative distribution function (CCDF) (Vassiliades et al., 2017) is another important metric in QD family that plots, for every reward threshold R on the horizontal axis, the fraction of policies in the archive whose returns meet or exceed that value. By doing so, the CCDF conveys both the overall quality and the behavioral diversity of the archived policies, while also illustrating how performance is distributed across the archive (Batra et al., 2023).

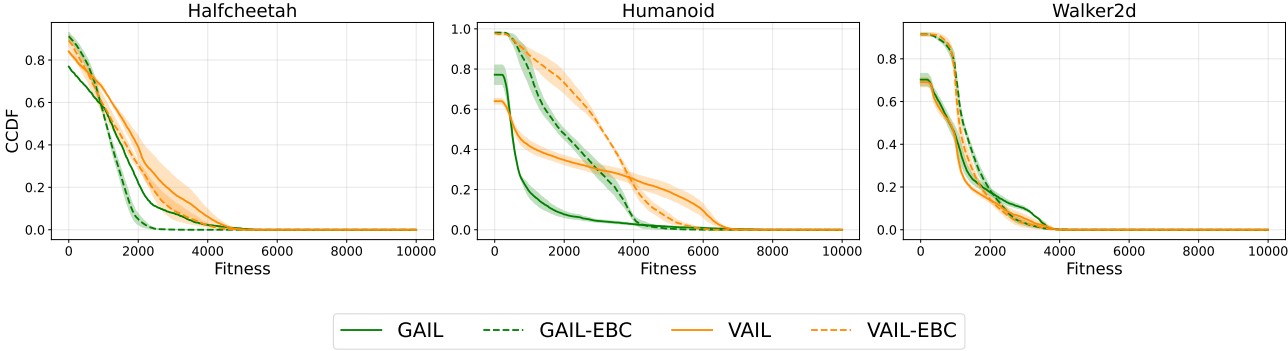

Figure 9: Plot of CCDF using GAIL, VAIL and their EBC variants across three environments.

As shown in Figure 9, we validate the effectiveness of EBC in imitation learning using GAIL and VAIL as baseline IRL models, and report the CCDF plot across three environments: Halfcheetah, Humanoid and Walker2d. Except for Halfcheetah, EBC consistently improves GAIL or VAIL with respect to the quality of learned policies, and we attribute this to the EBC mechanism's capability to unlock more potentially high-performing policies.

## G. QD Metrics Performance Results

Table 8 summarizes the performance of the different algorithms on the QD metrics.

Table 8: Quantitative performance comparison of QD-IRL methods (GAIL, VAIL, DiffAIL), their EBC-enhanced versions, PPGA with true reward, and three QD-IRL baselines. The mean performance across multiple random seeds is reported, with standard deviation indicated by $\pm$.

| | Halfcheetah | | | | Walker2d | | | | Humanoid | | | |
|---|---|---|---|---|---|---|---|---|---|---|---|---|
| | QD-Score | Cov(%) | Best | Avg | QD-Score | Cov(%) | Best | Avg | QD-Score | Cov(%) | Best | Avg |
| TrueReward | $6.54 \times 10^6 \pm 1.47 \times 10^5$ | $94.68 \pm 1.11$ | $8,724 \pm 384$ | $2,764 \pm 72$ | $3.66 \times 10^6 \pm 3.20 \times 10^4$ | $77.88 \pm 1.19$ | $5,454 \pm 189$ | $1,882 \pm 12$ | $7.38 \times 10^6 \pm 2.37 \times 10^6$ | $62.13 \pm 17.22$ | $9,636 \pm 78$ | $4,700 \pm 185$ |
| DiffAIL-EBC (Ours) | $3.99 \times 10^6 \pm 3.11 \times 10^5$ | $100.0 \pm 0.0$ | $4,290 \pm 117$ | $1,598 \pm 125$ | $2.93 \times 10^6 \pm 7.59 \times 10^4$ | $88.63 \pm 1.06$ | $2,283 \pm 109$ | $1,320 \pm 23$ | $8.92 \times 10^6 \pm 6.60 \times 10^5$ | $93.29 \pm 0.41$ | $5,598 \pm 298$ | $3,824 \pm 268$ |
| DiffAIL | $4.02 \times 10^6 \pm 5.82 \times 10^4$ | $99.72 \pm 0.28$ | $3,646 \pm 41$ | $1,614 \pm 28$ | $1.71 \times 10^6 \pm 4.08 \times 10^5$ | $52.14 \pm 0.74$ | $1,989 \pm 646$ | $1,304 \pm 294$ | $3.56 \times 10^6 \pm 3.71 \times 10^5$ | $76.32 \pm 0.48$ | $5,151 \pm 132$ | $1,869 \pm 206$ |
| VAIL-EBC (Ours) | $3.78 \times 10^6 \pm 7.69 \times 10^4$ | $99.14 \pm 0.50$ | $3,901 \pm 840$ | $1,526 \pm 39$ | $3.19 \times 10^6 \pm 2.15 \times 10^5$ | $91.45 \pm 1.62$ | $3,802 \pm 375$ | $1,398 \pm 72$ | $7.26 \times 10^6 \pm 3.10 \times 10^5$ | $97.63 \pm 0.59$ | $5,824 \pm 489$ | $2,975 \pm 145$ |
| VAIL | $3.62 \times 10^6 \pm 4.00 \times 10^4$ | $91.68 \pm 3.80$ | $5,123 \pm 401$ | $1,582 \pm 83$ | $2.40 \times 10^6 \pm 2.13 \times 10^5$ | $71.40 \pm 2.70$ | $3,570 \pm 417$ | $1,341 \pm 73$ | $5.09 \times 10^6 \pm 6.86 \times 10^5$ | $65.61 \pm 2.77$ | $7,056 \pm 250$ | $3,094 \pm 304$ |
| GAIL-EBC (Ours) | $2.64 \times 10^6 \pm 9.21 \times 10^4$ | $98.13 \pm 1.69$ | $2,770 \pm 440$ | $1,074 \pm 25$ | $3.42 \times 10^6 \pm 1.36 \times 10^5$ | $91.49 \pm 0.98$ | $4,022 \pm 569$ | $1,493 \pm 48$ | $1.86 \times 10^6 \pm 4.51 \times 10^5$ | $98.00 \pm 0.23$ | $5,140 \pm 852$ | $2,166 \pm 241$ |
| GAIL | $2.02 \times 10^6 \pm 8.36 \times 10^5$ | $67.83 \pm 16.05$ | $5,115 \pm 218$ | $1,165 \pm 341$ | $2.47 \times 10^6 \pm 2.88 \times 10^5$ | $69.27 \pm 4.51$ | $4,031 \pm 187$ | $1,425 \pm 73$ | $1.86 \times 10^6 \pm 4.51 \times 10^5$ | $82.25 \pm 9.30$ | $6,278 \pm 2,245$ | $924 \pm 251$ |
| GIRIL | $2.15 \times 10^6 \pm 8.98 \times 10^5$ | $95.96 \pm 0.79$ | $3,466 \pm 1,018$ | $909 \pm 382$ | $5.20 \times 10^5 \pm 1.33 \times 10^5$ | $25.08 \pm 3.93$ | $1,139 \pm 25$ | $821 \pm 112$ | $4.33 \times 10^6 \pm 2.63 \times 10^5$ | $67.32 \pm 6.16$ | $6,992 \pm 915$ | $2,588 \pm 253$ |
| AIRL | $3.11 \times 10^6 \pm 1.72 \times 10^5$ | $83.57 \pm 9.65$ | $5,183 \pm 1,735$ | $1,408 \pm 700$ | $2.53 \times 10^6 \pm 2.56 \times 10^5$ | $70.53 \pm 2.30$ | $4,280 \pm 326$ | $1,437 \pm 138$ | $2.31 \times 10^6 \pm 1.15 \times 10^5$ | $71.47 \pm 7.59$ | $7,661 \pm 705$ | $1,308 \pm 144$ |
| Max-Ent IRL | $1.12 \times 10^6 \pm 3.54 \times 10^5$ | $85.48 \pm 10.65$ | $2,570 \pm 710$ | $523 \pm 152$ | $1.80 \times 10^6 \pm 5.17 \times 10^4$ | $68.81 \pm 0.75$ | $3,756 \pm 318$ | $1,046 \pm 29$ | $1.82 \times 10^6 \pm 3.09 \times 10^5$ | $83.04 \pm 6.07$ | $4,658 \pm 1,489$ | $882 \pm 192$ |

## H. Tukey HSD Test Results

Table 9 compares our EBC-enhanced algorithms to their counterparts, to estimate the statistical significance of the results in Table 8. The test results show that 7/9 effects are positive and 5 of these are significant with $p \leq 0.05$.

Table 9: Tukey HSD test of the QD Score for the QD-IRL comparisons (w/o vs w/ EBC) in Table 8. $+$ indicates a positive effect of EBC. The significant pairwise comparisons ($p$<=0.05) are boldfaced.

| group1 | group2 | HalfCheetah | | Walker2d | | Humanoid | |
|--------|--------|:-----------:|---|:--------:|---|:--------:|---|
| DiffAIL | DiffAIL-EBC | - | 1.0 | + | **0.0236** | + | **0.0** |
| VAIL | VAIL-EBC | - | 0.9994 | + | 0.1894 | + | **0.0061** |
| GAIL | GAIL-EBC | + | 1.0 | + | **0.0231** | + | **0.0004** |

## I. Effects of Number and Quality of Demonstrations

Table 10 shows the results of our study on number and quality of demonstrations.

Table 10: Effects of demonstration number, diversity and quality on GAIL and GAIL-EBC in Humanoid. Values are mean $\pm$ one standard deviation over three seeds.

| Demos | Models | QD-Score | Coverage (%) |
|:-----:|:------:|:--------:|:------------:|
| 10 | GAIL | $2\,576\,765 \pm 82\,806$ | $68.68 \pm 3.20$ |
| | GAIL–EBC | $\mathbf{5\,822\,582} \pm 254\,060$ | $\mathbf{97.00} \pm 0.16$ |
| 4 | GAIL | $1\,886\,725 \pm 551\,004$ | $88.34 \pm 4.30$ |
| | GAIL–EBC | $\mathbf{5\,704\,650} \pm 150\,716$ | $\mathbf{97.84} \pm 0.04$ |
| 2 | GAIL | $2\,676\,218 \pm 360\,663$ | $70.44 \pm 0.20$ |
| | GAIL–EBC | $\mathbf{5\,803\,908} \pm 1\,320\,888$ | $\mathbf{97.30} \pm 0.30$ |
| 1 | GAIL | $1\,718\,577 \pm 134\,816$ | $75.24 \pm 1.12$ |
| | GAIL–EBC | $\mathbf{4\,948\,921} \pm 203\,595$ | $\mathbf{98.70} \pm 1.02$ |
| *Non-diverse (top-4) demos* | | | |
| | GAIL | $2\,832\,078 \pm 319\,164$ | $75.30 \pm 7.90$ |
| | GAIL–EBC | $\mathbf{4\,074\,729} \pm 306\,686$ | $\mathbf{98.12} \pm 0.40$ |

## J. Few-Shot Adaptation

For all adaptation tasks, the reward stays the same but the dynamics of the MDP is changed. The goal is to leverage the diversity of skills to adapt to unforeseen situations. Table 11 summarizes the details of the three adaptation tasks.

For all few-shot adaptation tasks, we evaluate all skills for each replication of each method and select the best one to solve the adaptation task.

On *Humanoid - Hurdles*, we use the jump features to jump over hurdles varying in height from 0 to 50 cm.

On *Humanoid - Motor Failure*, we use the feet contact features to find the best way to continue walking forward despite the damage. In this experiment, we scale the action corresponding to the torque of the left knee (actuator 10) by the damage strength ranging from 0.0 (no damage) to 1.0 (maximal damage).

On *Walker2d - Friction*, we use the feet contact features to find the best way to continue walking forward despite the change in friction. In this experiment, we scale the friction by a coefficient ranging from 0.0 (low friction) to 5.0 (high friction).

Table 11: Adaptation tasks

| Perturbations Range | Hurdles
height $\in [0, 50]$ | Motor Failure
damage strength $\in [0.0, 1.0]$ | Friction
coefficient $\in [0.0, 5.0]$ |
|---|---|---|---|
| MuJoCo Task | Humanoid | Humanoid | Walker2d |
| |  |  |  |
| Measures | Jump | Feet Contact | Feet Contact |

## K. Limitations

In this section, we discuss two potential limitations of our work that we have identified.

- The imitation reward functions learned by GAIL and VAIL are dynamically updated, and therefore it is still challenging to update the archives based on the imitation rewards since this error will accumulate over time. While we believe that the mechanism from MAP-Elites Annealing (Algorithm 2) alleviates the issue, we still observe significant performance declines in preliminary experiments and suggest future work is needed to ensure high-quality archives can be evolved.

- Due to the MDP property of QD-RL and QD-IRL and the mechanism of policy gradient, we can guide the search policy of PPGA towards the empty region of policy archive via the simple addition of curiosity-driven reward bonus. This is because the policy gradient methods will generate one gradient term which serves as improving the probability for the new policy to occupy empty cells. However, if the problem is not MDP-based or the policy optimization is not based on policy gradient, then our work may be not directly applicable.

## L. Discussion and Future Work

We first note the wide applicability of EBC. The measure **m** defining the behavior diversity of EBC is flexible, since it can be either pre-defined (as e.g. in the feet contact and angle measures in the paper) or based on a learning representation. As shown in our adaptation experiments, searching in this measure space can help to recover reasonable performance levels when the surface changes, the agent's limbs are damaged, and when new obstacles emerge. On the other hand, the measure could also be dynamically learned. For example, QDHF (Ding et al., 2024) adopts contrastive learning to learn the measure that best fits real-world human preferences. This fits particularly well in the extrinsic view on curiosity, where a human agent can serve as a critic of the behavioral diversity. In both cases, EBC could be applied to improve the efficiency and performance of QD-RL and QD-IRL algorithms, providing more promising application landscape. Moreover, our framework can potentially be extended to other robot locomotion tasks besides MuJoCo, which forms our future research plan.

For the algorithm design aspect, it is true that there are existing methods which intend to address the lack-of-exploration issue in QD context, such as CMA-ME (Fontaine et al., 2020). However, EBC allows a policy gradient algorithm to update the policy towards empty behavioral areas directly via incorporating the exploration incentive into gradients, which is more effective compared with the exploration of CMA-ME which doesn't utilize the gradient information. Moreover, we adopt a naive yet straightforward strategy to propagate the episode-based reward bonus: we simply add it uniformly to the reward at each timestep, thereby crediting the action of every step in the episode. Designing a more principled propagation method is a promising direction for further improving the performance of EBC. Furthermore, imitation tasks grounded in real-world datasets (e.g., human motion capture data) represent a compelling application domain for our method, with this direction emerging as a priority for future research.

