# OpenReview forum: "Diversifying Policy Behaviors with Extrinsic Behavioral Curiosity"
_ICML.cc/2025/Conference — ICML 2025 poster_

### Official Review · Reviewer_tEMb · 2025-03-12

**Overall Recommendation:** 4

**Summary:**

This paper introduces Quality Diversity Inverse Reinforcement Learning (QD-IRL), a framework that integrates quality-diversity (QD) optimization with inverse reinforcement learning (IRL) to enable robots to learn diverse locomotion behaviors from limited demonstrations. The key innovation is Extrinsic Behavioral Curiosity (EBC), which rewards agents for discovering novel behaviors, encouraging a broader exploration of the behavior space. The proposed method is tested on multiple robot locomotion tasks and and improves the performance of some QD-IRL instances. The results show that EBC can surpass even expert performance in some cases.

**Claims And Evidence:**

The claims made in the paper are well-supported by experimental results, particularly across three benchmark environments (Halfcheetah, Walker2d, Humanoid).

**Essential References Not Discussed:**

The related work is well discussed.

**Experimental Designs Or Analyses:**

The experimental setup is robust. Clear comparisons showing improvements with EBC. However, more complex imitation tasks can enhance the expressiveness of the method.

**Methods And Evaluation Criteria:**

The proposed method is well-justified for the problem of diverse behavior generation in robotics. The criteria are appropriate and comprehensive for evaluating diverse locomotion behaviors.

**Other Comments Or Suggestions:**

- Important concepts should be briefly introduced when first mentioned, such as MAP-Elites, VPPO, etc.
- The measure function m defined in the experiments does not seem to reflect curiosity-driven exploration.
- The experiments were conducted only in a limited set of MuJoCo environments. Imitation tasks based on real-world data (e.g., human motion capture data) could better highlight the advantages of the proposed method.

**Other Strengths And Weaknesses:**

- Strengths
  - Novel combination of IL and QD for diverse behavior generation.
  - Extrinsic Behavioral Curiosity (EBC) is a simple yet effective exploration mechanism.
- Weaknesses
  - The experiments in the paper need to be further strengthened.
  - Source code not provided.

**Questions For Authors:**

- How is a sufficiently explored region defined?
- From the perspective of imitation learning theory, how can the learned policy outperform the expert policy?
- Is there an optimal balance between imitation rewards and EBC rewards?
- What happens if the expert demonstrations are of low quality?
- In Figure 2, the improvement in reward for EBC in the Halfcheetah and Walker2d tasks does not seem significant. Is this related to the difficulty of the tasks?
- Some more concise mutual-information-based methods[1-5] also achieve diverse behavioral policies. What are the advantages of QD-IRL compared to these approaches?

[1] Eysenbach B, Gupta A, Ibarz J, et al. Diversity is all you need: Learning skills without a reward function[J]. arXiv preprint arXiv:1802.06070, 2018.

[2] Li Y, Song J, Ermon S. Infogail: Interpretable imitation learning from visual demonstrations[J]. Advances in neural information processing systems, 2017.

[3] Strouse D J, Baumli K, Warde-Farley D, et al. Learning more skills through optimistic exploration[J]. arXiv preprint arXiv:2107.14226, 2021.

[4] Peng X B, Guo Y, Halper L, et al. Ase: Large-scale reusable adversarial skill embeddings for physically simulated characters[J]. ACM Transactions On Graphics (TOG), 2022.

[5] Fu H, Tang K, Lu Y, et al. Ess-InfoGAIL: Semi-supervised imitation learning from imbalanced demonstrations[J]. Advances in Neural Information Processing Systems, 2023.

**Relation To Broader Scientific Literature:**

The paper is well-positioned within the fields of Imitation Learning (IL), Inverse Reinforcement Learning (IRL), and Quality Diversity (QD) Optimization.

**Theoretical Claims:**

The theoretical foundation is solid, particularly:
- The derivation of the EBC reward bonus using an indicator function for novel behaviors.
- Lemma 3.1, which provides a probabilistic guarantee that EBC increases the likelihood of discovering new behaviors.

The proofs appear sound, and the methodology aligns well with existing QD and IRL literature.

---

> ### Author Rebuttal · Authors · 2025-03-31
>
> ## We feel gratitude for the valuable insights and suggestions, and below is our detailed response.
>
> - - -
> **Q1**: More complex imitation tasks can enhance the expressiveness of the method.
>
> **A1**: We have added experiments with additional tasks and measure. Please refer to our response to Q3 of reviewer Cxuk.
>
> - - -
> **Q2**: It would be helpful to add sensitivity studies on hyperparameters.
>
> **A2**: We have conducted a hyperparameter study for the $q$ hyperparameter, the scale of the EBC reward; these results are mentioned in Appendix E.4. Other hyperparameters follow the settings in original implementations since these are not newly introduced by our method.
>
> - - -
> **Q3**: Source code not provided.
>
> **A3**: We will open-source the code and seeds when the paper is published.
>
> - - -
> **Q4**: Important concepts should be briefly introduced, e.g. MAP-Elites, VPPO, etc.
>
> **A4**: The details of MAP-Elites can be found in Section 2.1. For VPPO, we only briefly mention it in Section 2.1 as Vectorized PPO in the context of PPGA and in Appendix A2 l.671-673. We will expand the explanation both in the main text and in the appendix.
> - - -
> **Q5**: The measure function **m** defined in the experiments does not seem to reflect curiosity-driven exploration.
>
> **A5**: The measure value itself is not the curiosity. Instead, whether or not the diverse measure has been visited represents the behavioral-level curiosity.
>
> - - -
> **Q6**: Imitation tasks based on real-world data (e.g., human motion capture data) could better highlight the advantages of the proposed method.
>
> **A6**. We have conducted additional experiments on various imitation learning locomotion tasks (see our response to Q3 of reviewer Cxuk), and we are happy to extend our method to imitation tasks based on real-world data in our future work.
>
> - - -
> **Q7**: How is a sufficiently explored region defined?
>
> **A7**: The authors see that the “sufficiently explored region” is mentioned on l.240 page 5. We will remove the “sufficiently” from the text. Basically, as soon as the region is visited, the EBC reward will be zero for consequent visits, to facilitate exploitation of high-performing policy in this region.
>
> - - -
> **Q8**: From the perspective of imitation learning theory, how can the learned policy outperform the expert policy?
>
> **A8**: When we talk about performance we typically talk about the QD score. Hypothetically, it is possible that the expert policy has a higher cumulative reward since the expert would not be the optimal policy. However, we do not claim that the learned policy outperforms the expert. In Humanoid (see Fig.2), we do observe a higher QD score and coverage compared to PPGA with true reward. However, as shown in Fig.4,  none of the techniques can outperform PPGA with true reward with EBC.
>
> - - -
> **Q9**: Is there an optimal balance between imitation rewards and EBC rewards?
>
> **A9**: We studied the scale of the EBC reward, $q$ in Appendix E.4. $q=2$ appears to be the best choice. A sufficiently high $q$ is needed to for the behavioral level exploration to encourage higher QD score. It is impossible to find the exact optimal choice since the q value is continuous and there are infinite choices, and the optimal choice varies from tasks to tasks, so we choose q=2 in our experiments.
>
> - - -
> **Q10**: What happens if the expert demonstrations are of low quality?
>
> **A10**: Because the demonstrations are selected to be diverse, the expert data are already sub-optimal. If the performance is much lower than even a local optimum solution, then it seems difficult to achieve policy improvement. However, this can then be said for all imitation learning methods since it is assumed that the demonstrations are of desirable quality (otherwise there is limited point to imitating them).
>
> - - -
> **Q11**: The improvement in the Halfcheetah and Walker2d tasks does not seem significant.
>
> **A11**. We have performed a **Tukey HSD Test** on Table 7; comparing our EBC-enhanced algorithm to their counterparts, and we find that 7/9 effects are positive and  5 of these are significant with $p \leq 0.05$. Also refer to our answer to Q4 of U4H2 for further  explanations.
>
> - - -
> **Q12**: Mutual-information-based methods also achieve diverse behavioral policies. What are the advantages of QD-IRL?
>
> **A12**: The mentioned references assume it is possible to sample skills from a skill distribution, and then condition policies based on the given skill variables. In our work, the measures are not known in advance and are determined from the evaluation. The key difference is that we can optimise diversity across a particular measure rather than be limited to a particular skill distribution. So typically, unless the distribution of said techniques is uniform across the entire feasible space, QD algorithms will have more emphasis on diversity. We will discuss this difference in the updated paper.
>
> - - -
> ## We hope our response fully address your concerns.

---

> > ### Comment · Reviewer_tEMb · 2025-04-09
> >
> > I would like to thank the authors for answering my questions and for providing additional experimental results. Although the task demonstration could be stronger, this work demonstrates novelty and potential. Therefore, I have revised my score to accept (4).

---

> > > ### Author Response · Authors · 2025-04-09
> > >
> > > The authors are pleased that your concerns have been addressed. Thank you again for your time and your effort to review our submission, as well as your valuable suggestions to improve our paper.

---

### Official Review · Reviewer_U4H2 · 2025-03-14

**Overall Recommendation:** 3

**Summary:**

This work proposes a new paradigm called Quality-Diversity Inverse Reinforcement Learning (QD-IRL) as well as a new component to encourage the acquisition of novel and diverse behaviors, called Extrinsic Behavioral Curiosity (EBC). The goal of the QD-IRL framework is to enable the agent to learn diverse and performant policies from limited demonstration data and can be integrated on top of existing imitation learning (IL) approaches. Thus, the authors can combine the best of both approaches, overcoming the problems of vanilla IL, which struggles to learn diverse behavioral patterns. This framework achieves that by using IRL to learn the reward function, and then it utilizes Differentiable QD, specifically, QD-RL, to optimize the solutions archive. The framework is evaluated on locomotion tasks based on the performance gains it offers when integrated into existing IL methods.

**Claims And Evidence:**

Most of the claims are supported by clear evidence. However, the proposed framework is quite generic, and thus, I believe the authors should have conducted experiments on more diverse tasks rather than focusing solely on locomotion. The selection of a more diverse set of tasks would have contributed to a more constructive evaluation.

**Essential References Not Discussed:**

The authors have provided all the essential references required for the reader to comprehend the proposed approach.

**Experimental Designs Or Analyses:**

I believe there is some vital information missing from the manuscript. The authors never refer to how they construct the archive for the QD algorithm. Additionally, there is not a single reference to what behavior descriptor they utilize for each experiment, so the reader can understand how solutions diversify. Lastly, even though the locomotion tasks are well-known, I think it is important for the authors to add more information about the tasks such as the policy's inputs.

**Methods And Evaluation Criteria:**

Aside from the comments above, which also apply here, I believe the evaluation criteria are on point. In general, for QD algorithms the coverage and QD score present a clear overview of the performance of such algorithms.

**Other Comments Or Suggestions:**

No other comments.

**Other Strengths And Weaknesses:**

One other strong point of the manuscript is that it is well written and most of the figures are very intuitive. On the other hand, it would have been very positive if the authors had provided some videos of the optimal solutions in the archive for the reader to understand how diverse the solutions are, since for such high-dimensional tasks, it is hard to illustrate the outcome of each solution in the archive.

**Questions For Authors:**

* Is the EBC values bounded?
* Could the authors explain how they did the visualization for Fig. 3 and elaborate a bit more on what this figure represents?

**Relation To Broader Scientific Literature:**

This work is novel enough, and it presents a very interesting extension of IRL to the QD optimization paradigm. Moreover, the fact that the proposed approach can be implemented on top of existing algorithms is very positive. Nonetheless, I am a little concerned about the results in Fig. 2. The general improvements in performance EBC has to offer, in some cases are very minimal. An example would be the Halfcheetah and Walker2d experiments, where the QD increase in most of the algorithms is not that significant.

**Theoretical Claims:**

No problems were found in the theoretical claims of the manuscript.

---

> ### Author Rebuttal · Authors · 2025-03-31
>
> ## We appreciate the valuable insights and suggestions, and below is our detailed response to address your concerns.
>
> - - -
> **Q1**: More diverse tasks other than locomotion will be beneficial.
>
> **A1**: While we limit the scope of the experiments to robot locomotion tasks, we do agree with the inclusion of much more diverse tasks to demonstrate the capability of our algorithm. In particular, we have extended our method to more diverse tasks and will add these contents in the revised paper. It is the same table also mentioned to reviewer Cxuk in Q3 (reproduced here for convenience):
>
>
> | Game      | Measure          | Model     | QD-Score              | Coverage           |
> |-----------|------------------|-----------|-----------------------|--------------------|
> | hopper    | jump            | GAIL      | 64017±5558            | 100.0±0.0          |
> | hopper    | jump             | GAIL-EBC  | 81987±6890            | 100.0±0.0          |
> | humanoid  | angle  | GAIL      | 63771±7113            | 53.0±3.0           |
> | humanoid  | angle   | GAIL-EBC  | 170273±19711          | 96.0±0.0           |
> | humanoid  | jump           | GAIL      | 18812±6412            | 41.0±31.0          |
> | humanoid  | jump             | GAIL-EBC  | 90631±61894           | 85.0±15.0          |
> | ant       | feet_contact     | GAIL      | -484468±3264          | 75.84±0.0          |
> | ant       | feet_contact     | GAIL-EBC  | -52521±383213         | 77.28±3.2          |
>
>
> - - -
> **Q2**: The authors never refer to how they construct the archive for the QD algorithm.
>
> **A2**: We use the archive construction rules based on MAP-Elites, which is mentioned in section 2.1.
>
> - - -
> **Q3**: It would be helpful to introduce the behavior descriptor for each experiment, and the details of each tasks.
>
> **A3** : The information about the behavior descriptor is referenced at various locations in the paper under the terminology “measure” (which is the terminology that was used in the PPGA paper). Specifically, the measure used is highlighted in Section 4 as follows:
>  “the measure function maps the policy into a vector where each dimension indicates the proportion of time a leg touches the ground.” Moreover, we have clarified definition of state (policy’s inputs) and action (policy’s outputs) of each task in our revised paper.
>
> - - -
> **Q4**: The general improvements in performance EBC has to offer, in some cases are very minimal.
>
> **A4**:  Actually, when comparing performance, we should compare the base method with this method plus EBC (e.g., GAIL vs. GAIL-EBC) in terms of the QD-score. In the QD-score plots, the authors can only observe 2 comparisons among 9 comparisons (3 pairs $\times$ 3 environments), namely the Halfcheetah for DiffAIL-EBC vs DiffAIL and VAIL-EBC vs VAIL, that are not significantly better.  We highlight that EBC improves the QD score of GAIL, VAIL, and DiffAIL by up to 185%, 42%, and 150%, and that the standard deviations are very small. We have performed a **Tukey HSD Test** on Table 7; comparing our EBC-enhanced algorithm to their counterparts, and we find that 7/9 effects are positive and 5 of these are significant with $p \leq 0.05$.
>
> - - -
> **Q5**: It would be helpful to provide videos to illustrate the learned diverse solutions.
>
> **A5**: Thanks for the suggestion. One of the benefits of diversity is that the archives can be used for adaptation to new environment. In the updated submission, we will include a video of adaptation to new environments which can demonstrate the types of behaviors in the evolved archives as well as how this diversity can be exploited.
>
> - - -
> **Q6**: Is the EBC values bounded?
>
> **A6**: Since the unscaled EBC reward is either 0 or 1, so  the scaled EBC rewards are bounded in $[0,q]$, where $q$ is the hyperparameter that controls the weight.
>
> - - -
> **Q7**: Could the authors explain how they did the visualization for Fig. 3 and elaborate a bit more on what this figure represents?
>
> **A7**: Fig.3 represents the overall QD performance of the archive using a heatmap. The colors represent the fitness levels (quality), while the spread across the two-dimensional measure/behavior space represents the diversity.
>
> - - -
> ## We sincerely hope these responses fully address your concerns.

---

### Official Review · Reviewer_Y5D4 · 2025-03-14

**Overall Recommendation:** 3

**Summary:**

This paper proposes to combine Quality Diversity (QD) algorithms with Inverse Reinforcement Learning (IRL) problems. The authors introduce Quality Diversity Inverse Reinforcement Learning (QD-IRL), a method that uses rewards estimated from demonstrations (via GAIL, VAIL, and DiffAIL) with the PPGA quality diversity algorithm. The key contribution is Extrinsic Behavioral Curiosity (EBC), a reward mechanism that encourages exploration in unvisited areas of the measure space. Their experiments on three MuJoCo locomotion tasks (Halfcheetah, Walker2d, and Humanoid) to show that EBC significantly improves the performance of QD-IRL methods. The authors also demonstrate that EBC can enhance standard PPGA when applied to the true reward function.

**Claims And Evidence:**

The primary claim about EBC improving QD-IRL methods is supported by the experimental results. The performance improvements across metrics like QD-score, coverage, and reward are clearly demonstrated in the figures and tables.

However, the claim that QD approaches help with "adaptation in changing environments" (line 46) and "unpredictable real-world scenarios" (line 14) lack supporting evidence.
The authors do not provide experiments showing how the learned diversity could be practically useful, e.g. for scenarios like damage adaptation or hierarchical control, as done in prior works [1,2].
Such experiments would be useful to know if the EBC mechanism can really make agents more resilient.

**Essential References Not Discussed:**

A notable omission is the connection to improvement emitter mechanisms from CMA-ME [6], which use a similar approach to encourage exploration in empty areas of the archive.

**Experimental Designs Or Analyses:**

The authors appropriately test their approach with three different IRL methods (GAIL, VAIL, and DiffAIL) with and without EBC and compare against true reward baselines.
The experiments effectively show that EBC improves the QD-Score and Coverage in the 3 environments under study.

It would also be worth mentioning that the BestReward of EBC variants takes longer to converge (which is normal, as PPGA now optimizes for $fitness+q\times EBC$).

The computational requirements seem overly demanding (48 hours on 4 A40 GPUs per experiment).

**Methods And Evaluation Criteria:**

The proposed methods and evaluation criteria are generally appropriate for the problem. The QD metrics (QD-Score, Coverage, Best Reward, Average Reward) are standard and suitable for evaluating quality diversity algorithms.

However, the evaluation would be stronger if it included more diverse environments beyond the three locomotion tasks that all share similar objectives (maximizing forward progress) and measures (foot contact patterns). Adding tasks with different properties, such as an Ant Omni task from [5], would better demonstrate the generalizability of the approach.

Also, there is problem with the method design. While GAIL and VAIL rewards are updated at every iteration, the fitnesses stored in the archive are not regularly updated (as acknowledged in the limitations section, lines 907-910). This means that solutions in the archive are evaluated based on potentially outdated reward functions. This design choice could lead to poor performance because the archive might contain solutions that were highly rated under an old reward function but would score poorly under the current one. For example, AURORA [8] regularly updates the descriptors of the solutions stored in the archive (as the measure functions slightly changes every iteration).

**Other Comments Or Suggestions:**

- Consider reporting CCDF plots [4] (also called "archive profiles", see [5, 2]) to better characterize the distribution of episode returns in the final policy archives.
- Line 45 2nd column, the reference is likely meant to be [7]
- The reference to differential quality diversity (DQD) appears twice in the bibliography
- The size of the markers in Figure 2 makes it difficult to read
- Lines 377-379 should clarify that the baselines are only detailed in the Appendix
- Line 279: absolute values appear to be missing around $c_0$​

**Other Strengths And Weaknesses:**

Strengths:

- The proposed EBC reward is clear, sound, and well-motivated
- The results effectively show improvements in coverage and QD-score

Weaknesses:

- The authors describe their work as a "framework" (line 97 and abstract), but it's only applied to a specific QD algorithm (PPGA) and relies heavily on operations specific to gradient arborescence approaches. Indeed, the introduced EBC is only used when “branching solutions” and when “walking the search policy”, which are two steps very specific to PPGA and other gradient arborescence approaches like CMA-MEGA. Given the strong reliance on PPGA and gradient arborescence methods, the authors might consider renaming the algorithm to MEGA-IRL or PPGA-IRL rather than presenting it as a general framework
- The lack of experiments demonstrating the practical utility of the learned diversity (e.g. damage adaptation, hierarchical control...)
- Figure 4 is redundant with Figure 2 - the additional variant could simply have been included in Figure 2
- policy archives in Figure 3 have inconsistent colorbars, making the results difficult to interpret.

**Questions For Authors:**

- Lemma 3.1: the MDP reward appears to be non-Markovian, as it depends on the full trajectory. Is that normal? How does this affect the theoretical guarantees of your approach?
- Each experiment takes 48 hours to run on 4 A40 GPUs (line 746). Why does it take so long to run? How much faster would it be to run with other QD algorithms like CMA-MEGA?
- While you justify the usage of Quality-Diversity algorithm by the fact that it "allows adaptation in changing environments" (line 46), you do not provide any experiments showing why the learned diversity is useful. Could you demonstrate how the repertoires you obtain can be used (e.g., for damage adaptation, hierarchical control) as done in previous works [1,2]?
- You describe your approach as a "framework" (line 97 and abstract), but it's only applied to PPGA and relies on steps specific to gradient arborescence methods. How would EBC be implemented in other QD algorithms that don't use the same branching and walking mechanisms?

[1] Chalumeau, Felix, et al. "Neuroevolution is a competitive alternative to reinforcement learning for skill discovery."

[2] Grillotti, Luca, et al. "Quality-diversity actor-critic: learning high-performing and diverse behaviors via value and successor features critics."

[3] Faldor, Maxence, et al. "Synergizing quality-diversity with descriptor-conditioned reinforcement learning."

[4] Batra, Sumeet, et al. "Proximal policy gradient arborescence for quality diversity reinforcement learning."

[5] Flageat, Manon, et al. "Benchmarking quality-diversity algorithms on neuroevolution for reinforcement learning."

[6] Fontaine, Matthew C., et al. "Covariance matrix adaptation for the rapid illumination of behavior space."

[7] Cully, Antoine, et al. "Robots that can adapt like animals."

[8] Grillotti, Luca, and Antoine Cully. "Unsupervised behavior discovery with quality-diversity optimization."

**Relation To Broader Scientific Literature:**

To the best of my knowledge, this paper is the first to apply quality diversity to an inverse reinforcement learning setting. The authors implement their approach using the recent PPGA algorithm and introduce the EBC reward to enhance exploration. However, as presented, the EBC reward seems limited to the Gradient Arborescence subfamily of QD approaches (like CMA-MEGA and PPGA) and may not be applicable to other QD algorithms such as MAP-Elites, PGA-ME, and DCRL [3].

**Theoretical Claims:**

While the proof of Lemma 3.1 appears to be correct, I find one Lemma assumption intriguing: the MDP reward appears to be non-Markovian as it depends on the full trajectory (episode measure).

---

> ### Author Rebuttal · Authors · 2025-04-01
>
> ## Thank you for the valuable suggestions. Here is our response to questions and concerns.
>
> - - -
> **Q1**. This paper lacks evidence support for the significance of QD  (e.g. the ability of adaptation).
>
> **A1**. The claim that QD approaches help with adaptation in changing environments is reasonable and supported by prior works. We do notice that on l.46, the given reference does not adequately reflect the adaptation scenario, so we provide a more suitable few references. While the claim is not central to our work, we perform few-shot adaptation tasks for Humanoid-hurdles, and find that GAIL-EBC significantly outperforms GAIL on the return, and is comparable to the original QDAC performance of Grillotti et al. :
>
> |   Algorithm  | height 0.026  |  height 0.053    |  height 0.079     |  height 0.105          |
> |-- |------|---|---|---|
> GAIL |864.37135$\pm$331 | 329$\pm$65 | 198$\pm$26 | 107 $\pm$ 65 |
> GAIL-EBC | 2742$\pm$2252 | 396$\pm$105 | 275$\pm$23 | 130$\pm$89 |
>
> - - -
> **Q2**. More diverse IL tasks are needed to demonstrate the generalizability of the method.
>
> **A2**. We have conducted additional experiments with diverse measures. Please refer to our response to Q3 of reviewer Cxuk for details.
>
> - - -
> **Q3**. While GAIL and VAIL rewards are updated at every iteration, the finesses stored in the archive are not regularly updated (as acknowledged in the limitations section, lines 907-910).
>
> **A3**. That's indeed the limitation of our work that we already acknowledged, and we are happy to address this issue in our future work. Moreover, Algorithm 2, which dynamically adjust the fitness scores, mitigates the impact of this problem by computing a running average of past elite fitnesses.
>
> - - -
> **Q4**. EBC reward is non-Markovian and depends on whole trajectory, which may affect the theoretical guarantee.
>
> **A4**. The combination of non-markovian reward and PPO are explored and backed in prior work. For example,  [1]  validated that episode-based reward structures, when properly decomposed into each step, enhance PPO’s convergence and sample efficiency. Hence, we safely assume the convergence guarantee of PPO in our Lemma 3.1. Empirically, we also validated the effectiveness of EBC reward.
>
> [1] Arjona-Medina, Jose A., et al. "Rudder: Return decomposition for delayed rewards." Advances in Neural Information Processing Systems 32 (2019).
>
> - - -
> **Q5**. BestReward of EBC variants takes longer to converge (which is normal, as PPGA now optimizes for fitness+q×EBC).
>
> **A5**. In general, the higher the coverage, the longer it will take to evolve the elite solutions for all the covered cells. We will mention this in the text on page 7, l.364, in the text about Fig.2.
>
> - - -
> **Q6**. The computational requirements seem overly demanding (48 hours on 4 A40 GPUs per experiment).
>
> **A6**.  We apologize that the text is not clearly written. We use 1 GPU per experiment but run multiple experiments at the same time. We will rewrite this part.
>
> - - -
> **Q7**. This method should not be named as a framework. As presented, the EBC reward seems limited to the Gradient Arborescence subfamily of QD approaches (like CMA-MEGA and PPGA) and may not be applicable to other QD algorithms.
>
> **A7**. This is a limitation that we have acknowledged in Appendix G. Please note that we have implemented the EBC reward on different imitation learning methods, showing why it is usefully conceived as a framework. While MAP-Elites does not have an RL problem, PGA-ME and DCRL seem compatible for our framework. For instance, the  EBC reward would allow to take a step in the direction of the policy which generates new measures. Given the wide range of experiments, we leave this to future work.
>
> - - -
> **Q8**. The connection of CMA-ME and EBC regarding the exploration should be discussed.
>
> **A8**. We claim that EBC allows a **policy gradient** algorithm to update the policy towards empty behavioral areas **directly**, which is more effective compared with the exploration of CMA-ME which doesn't utilize the gradient information. For more details, please refer to Q2 from Reviewer Cxuk.
>
> - - -
> **Q9**. Figure 4 is redundant with Figure 2 - the additional variant could simply be included in Figure 2.
>
> **A9**. Figure 4 is for more straightforward comparison to show specifically that EBC can improve PPGA with true reward on conditions without EBC. We separated these so it is more easy to discuss in dedicated paragraphs in the text.
>
> - - -
> **Q10**. Policy archives in Figure 3 have inconsistent colorbars.
>
> **A10**. Please note that Figure 3 is plotted only for comparing the coverage metric, not the fitness (l.392). Therefore, the color (which represents fitness) is not our focus.
>
> - - -
> **Q11**. Consider reporting CCDF plots to better characterize the distribution of episode returns in the final policy archives.
>
> **A11**: Thanks for suggestion. We will report the CCDF plot in our revised paper.
>
> - - -
> ## We sincerely hope these responses fully address your concerns.

---

> > ### Comment · Reviewer_Y5D4 · 2025-04-08
> >
> > Thank you very much for the detailed answers, these address most of my concerns. I have updated my score for this paper. Depending on how the authors address the remaining issues, I may adjust my evaluation again before final decision.
> >
> > ----
> >
> > There is still one major point I disagree on:
> >
> > > A7: […] While MAP-Elites does not have an RL problem, PGA-ME and DCRL seem compatible for our framework. For instance, the EBC reward would allow to take a step in the direction of the policy which generates new measures. […]
> >
> > I agree the EBC reward can technically be integrated to PGA-ME and DCRL. However, the way it would be integrated to PGA-ME and DCRL is significantly different than from PPGA. In PGA-ME and DCRL, this reward would be added to the **emitter mechanism**, whereas in this work, it is added to the **gradient arborescence** mechanism. If you plan on adding it to the emitter mechanism, then you also need to compare to the improvement emitter from CMA-ME which has a similar mechanism.
> >
> > I do not think you need to compare to PGA-ME, DCRL, or CMA-ME. However, the claimed contribution, currently listed as EBC “can also significantly improve existing SOTA QD-RL algorithm” (line 107-108), is **too strong** compared to what you do in the rest of the paper, as you **only apply EBC to the gradient arborescence mechanisms of a specific algorithm**.
> >
> > I believe **your contribution is valuable**, but would be more accurately described as: *you introduce an EBC reward that improves exploration the Gradient Arborescence QD algorithms; and you show that when tested on the SOTA gradient arborescence algorithm PPGA, it enhances its exploration capabilities.*
> >
> > ----
> >
> > Other comments:
> >
> > > We claim that EBC allows a policy gradient algorithm to update the policy towards empty behavioral areas directly, which is more effective compared with the exploration of CMA-ME which doesn't utilize the gradient information.
> >
> > I mentioned CMA-ME in my initial review because the improvement emitter in CMA-ME already contains a mechanism to explicitly maximize behavioral diversity. While I acknowledge this mechanism differs from your proposed EBC reward, and **I don't believe you need to conduct direct comparisons with CMA-ME**, it's important to acknowledge this related prior work in your Related Work section. Your paper would be strengthened by explicitly discussing how your EBC approach relates to and differs from these existing behavioral diversity techniques.
> >
> > > That's indeed the limitation of our work that we already acknowledged, and we are happy to address this issue in our future work. Moreover, Algorithm 2, which dynamically adjust the fitness scores, mitigates the impact of this problem by computing a running average of past elite fitnesses.
> >
> > Thank you very much for the clarification. Indeed, the mechanism from MAP-Elites Annealing probably alleviates this issue. I think this limitation, together with your provided explanation from this rebuttal, should appear in the main paper (and not in appendix) as they are both quite important.

---

> > > ### Author Response · Authors · 2025-04-08
> > >
> > > The authors are pleased to see that most of the concerns have been addressed. Below you can find our response to the remaining issues.
> > >
> > > **Q1.** EBC in QD vs Arborescence QD algorithms.
> > >
> > > **A1.** The authors agree that the current framing of the contribution is too general given that we do not have supporting experiments. Therefore, we will rewrite where needed in the text that the EBC reward improves exploration in the context of Gradient Arborescence QD algorithms (rather than QD-RL algorithms in general).
> > >
> > > **Q2.** Your paper would be strengthened by explicitly discussing how your EBC approach relates to and differs from these existing behavioral diversity techniques.
> > >
> > > **A2.** The authors agree that a more detailed comparison would be beneficial, and particularly note that there is not sufficient discussion of related works about behavioral diversity methods in the context of QD. We will therefore improve the related work section accordingly.
> > >
> > > **Q3.** Clarification on the limitation of fitness computation.
> > >
> > > **A3.** The authors are pleased that the explanation of how we use Algorithm 2 has clarified that we do alleviate the problem. Indeed, the authors agree that it is best to highlight the limitation as well as how Algorithm 2 can alleviate the problem in the main text. So the authors will update the paper accordingly, along with suitable background references to related problems and solutions where needed.
> > >
> > > Thanks again for your valuable suggestion and we hope your concerns are fully addressed.

---

### Official Review · Reviewer_Cxuk · 2025-03-18

**Overall Recommendation:** 3

**Summary:**

Existing imitation learning algorithms fail to learn diverse behaviors. To address this, the paper introduces the QD-IRL framework that applies QD optimization algorithms to IRL problems. To further improve the exploration of QD-IRL, the paper introduces Extrinsic Behavioral Curiosity (EBC) to encourage policies to explore areas that are not covered. Experimental results show that EBC achieves better performance.

**Claims And Evidence:**

The claim is overall clear. However, similar frameworks seem to have been proposed in existing papers, e.g., Yu et al. (2024) cited in the paper. It would be better to have a discussion about the relationship.

The claim of the limitation of PPGA is not convincing to me. The paper claims that *the fitness term $f$ heavily influences PPGA’s search policy update direction* and *PPGA frequently becomes stuck in local regions*. However, as is shown in Eq. (2) in the paper, PPGA optimizes a weighted sum of fitness term $f$ and behavior measure terms $m_j$. The weights $c$ are optimized by CMA-ES, which tries to find the best weights to maximize the QD-Score improvement. Therefore, as the QD-Score considers both quality and diversity, PPGA is expected to adapt the weights $c$ dynamically to encourage the policies to explore the areas with low quality or no policy to improve QD-Score, rather than heavily optimizing the quality. Thus, PPGA does not seem to have these issues. It is not fully clear to me why EBC performs well.

**Essential References Not Discussed:**

NA

**Experimental Designs Or Analyses:**

- The experiments are not sufficient. The methods are only evaluated on three MuJoCo tasks with the same measure function (i.e., the proportion of time each leg touches the ground) on the same dimension (2). What are the performances on the tasks with higher or lower dimensions of measures (e.g., Ant and Hopper)? What are the performances on the tasks with other types of measure functions?
- The impact of the demonstrations is not analyzed. It may be critical to the performance of the method. What is the impact of the number of the demonstrations? Do they need to be diverse? What is the impact of the quality of the demonstrations?
- The advantages of the QD algorithms over the classical IRL algorithms are not analyzed. It would be helpful to add the pure IRL algorithms (without PPGA) with and without EBC bonus as baselines as well.
- I am glad that the QD metric values and their standard deviations are presented in Table 7 in the appendix. However, as each experiment was only conducted with 3 seeds, the standard deviations (and the averaged performance) in both Figure 2 in the body and Table 7 in the appendix may not be valid. It would be helpful to run more (>4) seeds and utilize the statistical tests to show the significance of the results.
- The seeds are not shown, and the code is not available currently, which may be harmful to the reproductivity of the results.

**Methods And Evaluation Criteria:**

The proposed method overall makes sense for the problem.

**Other Comments Or Suggestions:**

The cross (x) markers in the figures are mixed up. It would be better to improve their presentation.

**Other Strengths And Weaknesses:**

NA

**Questions For Authors:**

- PPGA also optimizes behavior diversity and aims to maximize the QD-Score. Why does PPGA with EBC achieve a better QD-Score than PPGA without EBC?
- What is the relationship with Yu et al. (2024) cited in the paper? It would be better to have a discussion.
- What are the performances on the tasks with other dimensions and other types of measure functions?
- What is the impact of the demonstrations?
- In Humanoid, DiffAIL-EBC even performs better than PPGA with true reward. Can you provide an analysis of it?
- Is it necessary to compare the average rewards of the methods? According to the definition, when a policy with a new behavior but a relatively low reward is added to the QD archive (which is what we intend to do), the average reward will fall instead. This conflicts with the goal of QD.
- Will the code and the seeds be open-sourced when the paper is published? This may be important for the reproducibility and validity of the experimental results.

I would be happy to raise my score if the concerns and questions are answered.

**Relation To Broader Scientific Literature:**

According to the results, the proposed EBC method may also improve the performance (coverage) of classical QD-RL algorithms.

**Theoretical Claims:**

The paper provides a slightly trivial lemma showing that optimizing the EBC reward helps cover more areas in the behavior space.

---

> ### Author Rebuttal · Authors · 2025-03-31
>
> ## Thanks for the valuable insights and suggestions.
> - - -
> **Q1.** It would be better to discuss the relationship between Yu et al.'s paper.
>
> **A1.** The key difference between Yu’s work and our work lies in two aspects:
> - Yu’s work adopts a ​**single-step** reward bonus (calculated per (s,a) pair), while our method uses ​**episode-based reward** bonus aligned with whole-episode measurements.
> - Yu’s work focuses on QD-IL, whereas our framework extends to QD-RL.
>
> We will add these discussions in our revised paper.
>
> - - -
> **Q2.** Why use EBC given CMA-ES already handles quality/diversity?
>
> **A2.** The EBC reward can supplement the PPGA algorithm: while PPGA works by evolving random coefficients for each measure dimension, the EBC reward encourages the policy to unlock new areas of measure space by adding it to the fitness, which is independent of the local measure space, for a more direct, global, and efficient search of new behavioral areas.
>
> - - -
> **Q3.** More experiment needed on tasks with different measure dimensions and type of measures.
>
> **A3.** We tested EBC across tasks with varying measure dimensions (**m**):
> - Jump (1D: lowest foot height)
> - Angle (2D: body angle)
> - Feet-contact (4D: leg-ground time)
>
> GAIL-EBC consistently outperforms GAIL in all scenarios:
>
> | Game      | Measure          | Model     | QD-Score              | Coverage           |
> |-----------|------------------|-----------|-----------------------|--------------------|
> | hopper    | jump            | GAIL      | 64017±5558            | 100.0±0.0          |
> | hopper    | jump             | GAIL-EBC  | 81987±6890            | 100.0±0.0          |
> | humanoid  | angle  | GAIL      | 63771±7113            | 53.0±3.0           |
> | humanoid  | angle   | GAIL-EBC  | 170273±19711          | 96.0±0.0           |
> | humanoid  | jump           | GAIL      | 18812±6412            | 41.0±31.0          |
> | humanoid  | jump             | GAIL-EBC  | 90631±61894           | 85.0±15.0          |
> | ant       | feet_contact     | GAIL      | -484468±3264          | 75.84±0.0          |
> | ant       | feet_contact     | GAIL-EBC  | -52521±383213         | 77.28±3.2          |
>
> - - -
> **Q4.** How do demonstration(demo) quantity/diversity/quality impact performance?
>
> **A4.** Experiments with varying demo counts in Humanoid environment:
>
> | Demos | Model     | QD-Score         | Coverage      |
> |-------|-----------|------------------|---------------|
> | 10    | GAIL      | 2576765±82806    | 68.68±3.2     |
> |       | GAIL-EBC  | 5822582±254060   | 97.0±0.16     |
> | 4     | GAIL      | 1886725±551004   | 88.34±4.3     |
> |       | GAIL-EBC  | 5704650±150716   | 97.84±0.04    |
> | 2     | GAIL      | 2676218±360663   | 70.44±0.2     |
> |       | GAIL-EBC  | 5803908±1320888  | 97.3±0.3      |
> | 1     | GAIL      | 1718577±134816   | 75.24±1.12    |
> |       | GAIL-EBC  | 4948921±203595   | 98.7±1.02     |
>
> **Key observations:**
> - With 1 demo, GAIL-EBC’s QD-Score drops by 15% vs. 10-demos.
> - Using non-diverse elites (top4 best-reward):
>
> | Algorithm     | QD-Score         | Coverage      |
> |----------------|--------|---------------|
> |     GAIL      |  2832078±319164    | 75.3±7.9   |
> |   GAIL-EBC  |  4074729±306686 (-30% vs 4-demos) | 98.12±0.4  |
>
> - - -
> **Q5**. It would be helpful to add the pure IRL algorithms (without PPGA) with and without EBC bonus as baselines.
>
> **A5**. Without the QD algorithm, there would be no QD-Score, coverage, and average reward metrics. So we would only be able to compare the maximal fitness, which is not the our objective.
>
> - - -
> **Q6**. It would be helpful to use statistical tests to show the significance of the results.
>
> **A6**. We perform a **Tukey HSD Test** on Table 7; comparing our EBC-enhanced algorithm to their counterparts, 7/9 effects are positive and 5 of these are significant with $p \leq 0.05$.
>
> - - -
> **Q7**. The code is not available.
>
> **A7**. We will open-source the code and the seeds when the paper is published.
>
> - - -
> **Q8**. PPGA also optimizes behavior diversity and aims to maximize the QD-Score. Why does PPGA with EBC achieve a better QD-Score than PPGA without EBC?
>
> **A8**. We kindly refer you to A2. PPGA itself is not sufficient in exploration. However, our EBC bonus synergized with CMA-ES could facilitate exploration and exploitation more effectively (see our paper line 244).
>
> - - -
> **Q9.** Why does DiffAIL-EBC outperform PPGA with true reward in Humanoid?
>
> **A9.** As shown in Fig.4, PPGA itself's performance improves significantly with EBC (improved exploration). However, DiffAIL-EBC does ​not surpass PPGA-EBC, aligning with expectations. See analysis in  l.420-428 (Page 8).
>
> - - -
> **Q10.** Is comparing average rewards necessary?
>
> **A10.** While QD-score is the primary metric, average reward provides insights into archive quality. We retain it for supplementary analysis but emphasize QD metrics as the core evaluation.
>
> - - -
> ## We hope these answers address your concerns.

---

### Decision · Program_Chairs · 2025-05-01

**Decision:**

Accept (poster)

**Comment:**

In this submission, the authors introduce Quality Diversity Inverse Reinforcement Learning (QD-IRL), a new framework which integrates quality-diversity (QD) optimization with inverse reinforcement learning (IRL). QD-IRL enables agents to learn diverse behaviours from limited demonstrations. The paper also introduces Extrinsic Behavioral Curiosity (EBC) which provides agents with additional rewards from an external critic based on the novelty of the agent’s behavior. The authors provide experimental results validating the effectiveness of EBC in exploring diverse locomotion policies and also compare it with existing baselines.

The reviewers believe that the EBC reward and the application of IRL to QD algorithms are novel and important contributions. Furthermore, they agree that the claims presented in the paper are sufficiently supported by empirical evidence. That said, reviewers also raised several concerns, including the lack of diverse tasks (other than locomotion) and the lack of results demonstrating practical utilities of learned behaviour (e.g. adaption or hierarchical control). The reviewers would also believe that including videos of optimal solutions in the archive would strengthen the paper, as it is difficult to understand how diverse the solutions are based on this. In the rebuttal, the authors have addressed many of the concerns, which has led to increases in scores.

Overall Assessment: Based on the reviewers' comments and discussions during and after the rebuttal, I believe that this paper makes a strong technical contribution and is worthy of being accepted at ICML. I hope the reviewers’ comments can help the authors prepare an improved version of this submission for camera-ready.